# TDP-43 facilitates milk lipid secretion by post-transcriptional regulation of *Btn1a1* and *Xdh*

Limin Zhao[1,2,12], Hao Ke[1,2,12], Haibo Xu [1,2], Guo-Dong Wang[1,3], Honglei Zhang [1], Li Zou[1], Shu Xiang[4], Mengyuan Li[5], Li Peng[6], Mingfang Zhou[5], Lingling Li[1,7], Lei Ao[8], Qin Yang[1], Che-Kun James Shen[9], Ping Yi[5]*, Lu Wang[10]* & Baowei Jiao [1,3,11]*

Milk lipid secretion is a critical process for the delivery of nutrition and energy from parent to offspring. However, the underlying molecular mechanism is less clear. Here we report that TDP-43, a RNA-binding protein, underwent positive selection in the mammalian lineage. Furthermore, TDP-43 gene (*Tardbp*) loss induces accumulation of large lipid droplets and severe lipid secretion deficiency in mammary epithelial cells to outside alveolar lumens, eventually resulting in lactation failure and pup starvation within three weeks postpartum. In human milk samples from lactating women, the expression levels of TDP-43 is positively correlated with higher milk output. Mechanistically, TDP-43 exerts post-transcriptional regulation of *Btn1a1* and *Xdh* mRNA stability, which are required for the secretion of lipid droplets from epithelial cells to the lumen. Taken together, our results highlights the critical role of TDP-43 in milk lipid secretion, providing a potential strategy for the screening and intervention of clinical lactation insufficiency.

[1] State Key Laboratory of Genetic Resources and Evolution, Kunming Institute of Zoology, Chinese Academy of Sciences, 650223 Kunming, China. [2] Kunming College of Life Science, University of Chinese Academy of Sciences, 650223 Kunming, China. [3] Center for Excellence in Animal Evolution and Genetics, Chinese Academy of Sciences, 650223 Kunming, China. [4] The First Hospital of Kunming, Calmette International Hospital, 650011 Kunming, China. [5] Department of Obstetrics and Gynecology, The Third Affiliated Hospital of Chongqing Medical University, 401120 Chongqing, China. [6] Yubei District Maternal and Child Health Care Hospital, 401120 Chongqing, China. [7] School of Life Sciences, University of Science and Technology of China, 230026 Hefei, China. [8] Kunming Angel Women's and Children's Hospital, 650032 Kunming, China. [9] Institute of Molecular Biology, Academia Sinica, 11529 Taipei, Nankang, Taiwan. [10] State Key Laboratory for Conservation and Utilization of Bio-resources in Yunnan, Yunnan University, 650091 Kunming, China. [11] KIZ-CUHK Joint Laboratory of Bioresources and Molecular Research in Common Diseases, Kunming Institute of Zoology, Chinese Academy of Sciences, 650223 Kunming, China. [12] These authors contributed equally: Limin Zhao, Hao Ke. *email: 2362953558@qq.com; wanglu@ynu.edu.cn; jiaobaowei@mail.kiz.ac.cn

Mammalian milk not only delivers nutrients to successfully support offspring but also provides sufficient immunoregulatory agents for antimicrobial protection and neonatal survival[1,2]. However, little attention has been devoted to characterize key genes involved in milk secretion in mammals. In humans, increasing evidence has demonstrated that breastfeeding is highly beneficial for infants, including reduction in the incidence of diarrhea and pneumonia[3,4], decreased risk of obesity[5], and increased immune system maturation[6]. Moreover, breastfeeding can protect mothers against cardiovascular disease, metabolic syndrome[7], and breast cancer[3]. Despite this, only about 20% of women maintain exclusive breastfeeding for 6 months[8], with lactation insufficiency being the most cited reason for this finding[9–11]. Therefore, understanding the milk secretion process to improve early diagnosis and lactation performance is an urgent issue.

Among the components of milk, lipids are remarkable sources of energy for offspring in most mammals[12], and successful lipid secretion is critical for newborn survival during lactation[13,14]. As major components of the milk lipid droplet (LD) membrane for lipid secretion, butyrophilin 1a1 (BTN, encoded by the *Btn1a1* gene) is important for milk lipid secretion and therefore neonatal survival[13,15,16]. Xanthine oxidoreductase (XOR, encoded by the *Xdh* gene) is also reported to modulate milk lipid secretion during lactation[17–19]. However, although some studies have found that XOR can be regulated by various factors at the transcriptional level[20–22], little attention has been paid to post-transcriptional regulators. Several reports have suggested that regulation of RNA stability or regulation at the post-transcriptional level may be the key to lactation activation[23,24]. RNA-binding proteins (RBPs) mediate key steps in post-transcriptional regulation of gene expression[25–27]. Therefore, identifying those RBPs that control the post-transcriptional expression of essential genes in lactation would be helpful for delineating the milk secretion process.

Here, we perform likelihood ratio tests of RBPs to screen the potential regulators of lactation and find that TDP-43 experiences positive selection in mammals. Furthermore, KO of *Tardbp* (TDP-43 gene) in mice results in LD secretion failure, and thereafter lactation failure and poor newborn survival. The clinical samples from lactating women further emphasize the substantial role of TDP-43 in milk secretion. For the underlying mechanism, we show that TDP-43 could bind to the 3′-untranslated regions (UTRs) of the *Btn1a1* and *Xdh* transcripts and thereby regulate their messenger RNA (mRNA) stability. Importantly, our findings highlight the crucial role of TDP-43 in milk lipid secretion.

## Results

**Positive selection of TDP-43 gene during mammalian evolution**. Lactation is a crucial physiological factor in mammalian survival. Despite some studies suggesting that post-transcriptional regulation of lactation is important[23,24], little information is currently available on the functional roles of regulators on lactation at the post-transcriptional level. As RBPs mediate key steps in the post-transcriptional regulation of gene expression[25–27], we focused on identifying those RBPs essential for the regulation of lactation. Considering that lactation is a highly characteristic feature of mammals compared with other species, we identified lactation-related genes by calculating positive selection signals in the ancestral branch of mammals, as positively selected genes in these branches may be associated with mammalian characteristics in comparison to those of other animals (e.g., fish, birds, reptiles). We therefore performed phylogenomic analysis of positive selection in 15 vertebrate genomes to identify candidate RBPs for lactation. Likelihood ratio tests (LRTs) were first employed to

identify genes under positive selection in the list of mammalian RBPs[28]. After stringent filtering, 60 one-to-one orthologous groups (OGs) of RBPs across 15 species were analyzed (Supplementary Fig. 1 and Supplementary Table 1). The LRTs from the branch-site model showed that, in the ancient branch of mammals, *TARDBP* and *SRSF9* underwent significant positive selection, with *p* values of 0.031 and 0.022 after false discovery rate correction, respectively. In addition, the d$N$/d$S$ values were 371.37 and 12.41, respectively (Table 1). The multiple alignments and conserved codons of *TARDBP* and *SRSF9* are shown in Supplementary Figs. 2 and 3, respectively, suggesting that *TARDBP* and *SRSF9* may play important roles in mammalian features compared with other animals.

**TDP-43 loss in mammary epithelium induces lactation failure.** As milk secretion is one of the most characteristic features of mammalian species, and TDP-43 (encoded by the *TARDBP* gene) is involved in breast cancer progression[29], we hypothesized that TDP-43 may play an important role in milk secretion, while the other candidate, *SRSF9*, was excluded for further investigation because of its negatively regulatory roles on milk secretion-related genes (BTN and XOR) (Supplementary Fig. 4A, B). The expression pattern of TDP-43 at different mammary gland stages was first examined in mice. Immunohistochemical assay showed that the TDP-43 protein was highly expressed during late pregnancy and early lactation in comparison with that during virginity and involution (Fig. 1a, ×40 objective magnification; and Supplementary Fig. 4C, ×10 objective magnification). These results were confirmed by quantitative real-time polymerase chain reaction (qRT-PCR, Supplementary Fig. 4D) and western blot assays (Supplementary Fig. 4E, F). TDP-43 was located in both the myoepithelial and luminal cell layers, as indicated by its colocalization with cytokeratin 14 and 18 (CK14 and CK18) (Supplementary Fig. 4G), respectively.

To examine the roles of TDP-43 in the mammary epithelium during pregnancy and lactation, we disrupted TDP-43 expression in *Tardbp* floxed mice[30] using transgenic mice expressing Cre-recombinase (Cre) driven by the whey acidic protein (WAP) promoter activated during middle pregnancy to lactation in luminal epithelial cells[31]. The WAP-Cre line abrogated TDP-43 expression effectively at pregnancy day 17.5 (P17.5) and at lactation day 10 (L10), confirming *Tardbp* knockout (KO) in the mammary epithelium (Supplementary Fig. 5A–C). To characterize the overall survival of offspring, we calculated the survival rate of pups from various litter sizes. First-litter pups born to *Tardbp* KO female mice (*Tardbp*[flox/flox]+WAP-Cre, shown as *Tardbp*[−/−] in the figures) showed obviously lower survival rates in comparison to those born to *Tardbp*-intact mice (WAP-Cre, shown as wild type (WT) in the figures) (Supplementary Fig. 5D). Although the pup survival rate improved in the KO group during the second lactation, the overall trend was consistent with that of the first gestation (Supplementary Fig. 5E). To exclude discrepancy in nourishment caused by different sized litters, the pup survival rate was analyzed for the same sized litters (6, 7, or 8 pups per litter) (Fig. 1b, c and Supplementary Fig. 5F–H), which showed a similar pattern of declining pup survival.

At the end of L2, we adjusted the litter size to seven to observe the weight of the surviving pups. Although most pups died before L2, those pups that survived lived until L13, after which time the survival rate markedly decreased from L15 to L23 (Fig. 1d). The survival rate using unadjusted litter sizes also exhibited a similar decline after L15 (Supplementary Fig. 5D). Furthermore, the average weights of the surviving pups born to *Tardbp*[−/−] mothers were substantially lower than those born to WT mothers during the first (Fig. 1e, black lines) and second (Fig. 1f) lactations. To

**Table 1 Positive selection on mammalian *TARDBP* and *SRSF9*.**

| Gene | lnL0 | lnL1 | 2ΔlnL | dN/dS | Positive selection site |
|---|---|---|---|---|---|
| *TARDBP* | −5443.244 | −5437.227 | 12.04* | 371.37 | 352P*, 360Q |
| *SRSF9* | −1846.979 | −1852.675 | 11.39* | 12.41 | 1N, 22V, 44R, 129R, 159R |

lnL log likelihood, lnL0 LnL under model A null hypothesis, lnL1 LnL under model A alternative hypothesis
*Five percent significance level

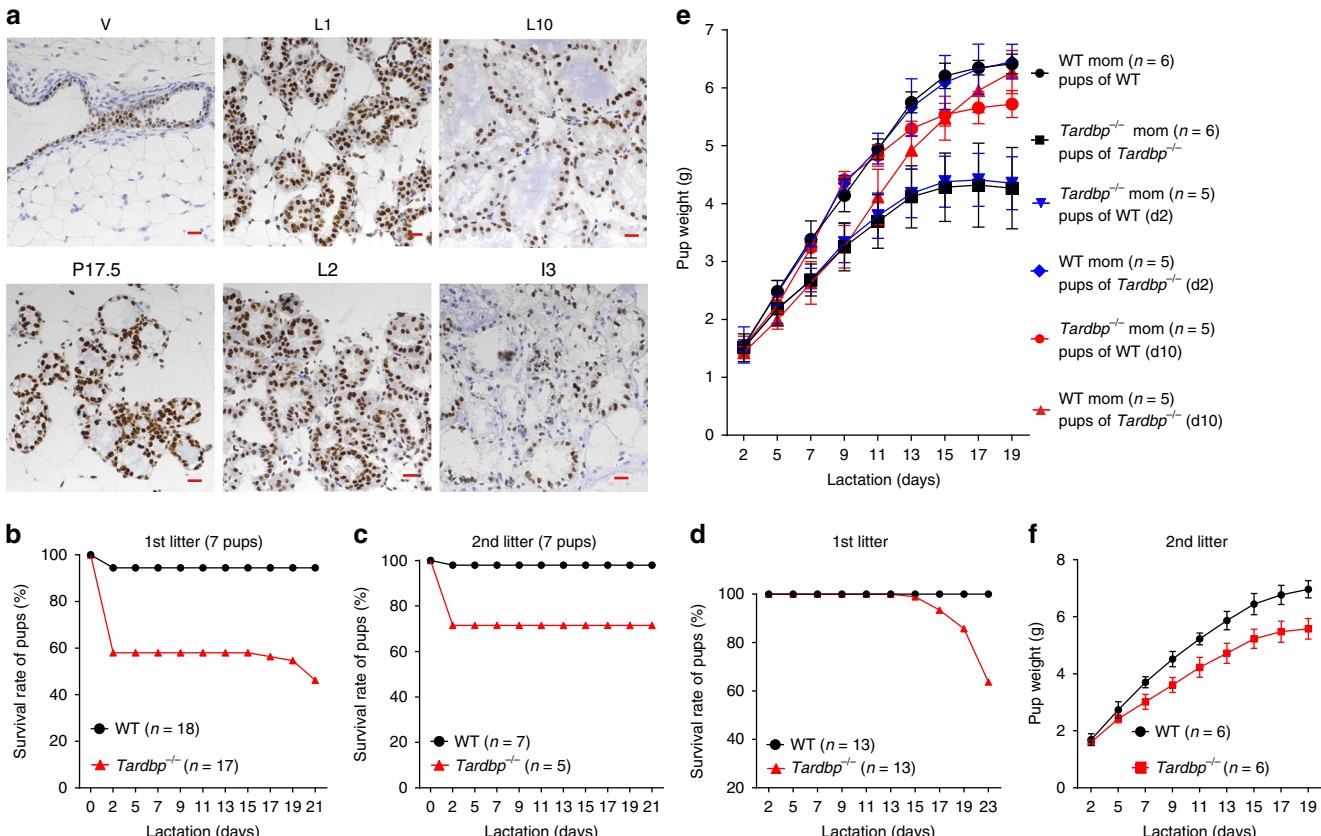

**Fig. 1 TDP-43 loss induces lactation failure in mammary epithelium. a** Immunohistochemical (IHC) staining of TDP-43 in mammary glands of C57BL/6 female mice at different developmental stages (V, virgin; P17.5, pregnancy day 17.5; L1, lactation day 1; L2, lactation day 2; L10, lactation day 10; I3, involution day 3). Representative images (×40 objective magnification) from four mice are shown for each stage. Scale bar: 20 μm. **b**, **c** Pup survival rates from wild-type (WT) and *Tardbp*⁻/⁻ female mice for first (**b**) and second (**c**) lactation, with seven pups per litter. WT represents *Tardbp*^wt/wt^+WAP-Cre mice. *Tardbp*⁻/⁻ represents *Tardbp*^fl/fl^+WAP-Cre mice. *n* represents number of mothers. **d** Pup survival rates from WT and *Tardbp*⁻/⁻ mothers after litter size was adjusted to seven for each foster mother on L2. **e** Body weights of surviving pups from WT and *Tardbp*⁻/⁻ mothers. Pups nursed by their biological mother (WT or *Tardbp*⁻/⁻, black lines) and cross-nursed by their foster mother (*Tardbp*⁻/⁻ or WT) at L2 (blue lines) and L10 (red lines) were recorded for each dam with seven pups beginning at L2. **f** Body weights of pups from second lactation after litter size was adjusted to seven on L2. Data are means ± SD. Source data are provided as a Source Data file.

exclude possible defects in pups, pups born to WT and *Tardbp*⁻/⁻ mothers were cross-fostered by *Tardbp*⁻/⁻ and WT mothers, respectively. Results demonstrated that pups born to *Tardbp*⁻/⁻ female mice but fostered by WT mice beginning at L2 grew with normal body weights and survival rates, whereas pups born to WT females but fostered by *Tardbp*⁻/⁻ mice exhibited reduced growth (Fig. 1e, blue lines and Supplementary Fig. 5I). A similar reduction in body weight was observed when pups born to WT females were fostered by *Tardbp*⁻/⁻ females at the beginning of L10 (Fig. 1e, red lines). These results indicate that TDP-43 is required for lactation to support offspring survival in mice. To ensure that the observed failure of offspring growth and viability was dependent on the *Tardbp* genotype of the mothers, rather than that of the pups themselves, we genotyped all deceased pups and found normal genotype distribution (Supplementary Fig. 5J). Overall,

these results demonstrate that maternal *Tardbp*⁻/⁻ induced failure of both pup viability and growth.

**TDP-43 loss results in disrupted lipid secretion.** Morphological changes were first evaluated to determine the possible causes of lactation defects in *Tardbp* KO mice. Mammary glands from *Tardbp*⁻/⁻ mice harvested at the mid-pregnancy (P15.5, P17.5) to lactation stages (L2 and L10) demonstrated no differences in either lobuloalveolar structures or alveolar densities compared with WT mice based on whole-mount (Supplementary Fig. 6A) and hematoxylin and eosin (H&E) staining (Supplementary Fig. 6B). Moreover, TUNEL assay and Ki67 staining revealed indistinguishable rates of apoptosis and proliferation at P17.5 (Supplementary Fig. 6C–F). These results suggest that pup deaths were unlikely due to structural defects caused by *Tardbp* KO.

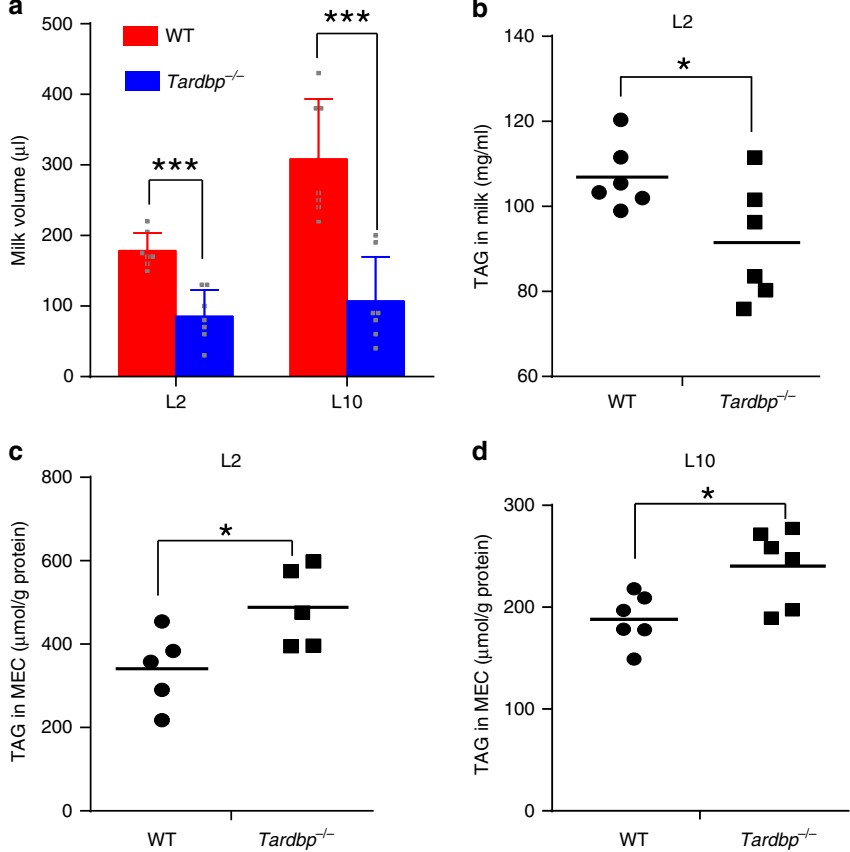

**Fig. 2 Milk secretion is impaired in _Tardbp_$^{-/-}$ female mice. a** Analysis of milk volume from mammary glands of wild-type (WT) and _Tardbp_$^{-/-}$ female mice following 10 U of oxytocin stimulation ($n = 7$ for each genotype). **b** Analysis of triacylglycerols (TAGs) in milk collected from WT and _Tardbp_$^{-/-}$ mice at lactation day 2 (L2) ($n = 6$ for each genotype). **c**, **d** Analysis of TAGs in mammary gland epithelial cells (MECs) collected from WT and _Tardbp_$^{-/-}$ mice at L2 (C) ($n = 5$ for each genotype) or L10 (D) ($n = 6$ for each genotype). Data are means ± SD (**a**) or means (**b**–**d**). TAG levels in MECs were normalized to protein concentration of each sample. Unpaired _t_ test was used to evaluate statistical significance. *$P < 0.05$, ***$P < 0.001$. Source data are provided as a Source Data file.

To identify factors contributing to the death of pups nursed by _Tardbp_ KO mice, we focused on milk, the main energy source for offspring survival. Milk was collected from lactating mice for volume and compositional analyses. _Tardbp_ KO led to a significant decrease in milk secretion compared to that in WT mice at L2 and L10 using two concentrations of oxytocin (the higher dose to remove all milk at an early stage and the lower dose at an established lactation in conventional use), respectively (Fig. 2a and Supplementary Fig. 7A). We next measured changes in milk protein and lipid compositions, which are the predominant sources of nutrition in milk. The protein quantitation assays demonstrated that the protein compositions of milk collected from lactating _Tardbp_$^{-/-}$ and WT mice at L2 were indistinguishable (Supplementary Fig. 7B, C), and the expression levels of several essential genes related to milk protein in mammary gland epithelial cells (MECs) were unaffected by _Tardbp_ loss (Supplementary Fig. 8A–C). However, the concentrations of triacylglycerols (TAGs), the predominant component of lipids, were markedly lower in milk taken from _Tardbp_$^{-/-}$ mice compared with that from WT mice (Fig. 2b). Polyunsaturated fatty acids and essential fatty acids, which are indicators of exogenous uptake, were similar between the milk from _Tardbp_$^{-/-}$ and WT mice (Supplementary Fig. 7D, E), suggesting that uptake by the mice was the same. These observations suggest that loss of TDP-43 can lead to the abnormal composition of milk lipids but not of milk proteins.

To further clarify lipid disorder induced by _Tardbp_ KO, we collected MECs to analyze their lipid concentrations. Interestingly, although TAG concentrations in milk were markedly lower in _Tardbp_$^{-/-}$ mice (Fig. 2b), the cellular TAG concentrations in MECs were much higher in _Tardbp_$^{-/-}$ mice than that in WT mice at L2 and L10, respectively (Fig. 2c, d). These results indicate that the phenotype observed in our study was likely due to unsuccessful milk lipid secretion from MECs to the outside, resulting in the accumulation of higher TAG concentration in MECs but lower TAG concentration in milk. Thus, these results indicate that lipid secretion was affected by _Tardbp_ KO.

**TDP-43 loss results in large LD accumulation.** Because milk lipids are secreted as milk fat globules (MFGs) wrapped in a bilayer plasma membrane, we compared MFGs between WT and _Tardbp_$^{-/-}$ mice. Results indicated that MFGs from _Tardbp_$^{-/-}$ mice were substantially larger than those from WT mice at L2 and L10 (Fig. 3a, b). Moreover, milk alveolar structures and MECs from _Tardbp_ KO mice were clearly filled with large cytoplasmic LDs (CLDs) or LDs during lactation (L1 to L10), whereas large LDs or CLDs were not found in WT mice (Fig. 3c). This phenomenon was maintained during the second lactation in KO mice (Supplementary Fig. 9A). To confirm the above H&E staining results, we conducted immunofluorescence staining of mammary glands from genotypes at L1, L2, and L10 using an antibody against PLIN2, a specific marker of milk LDs during

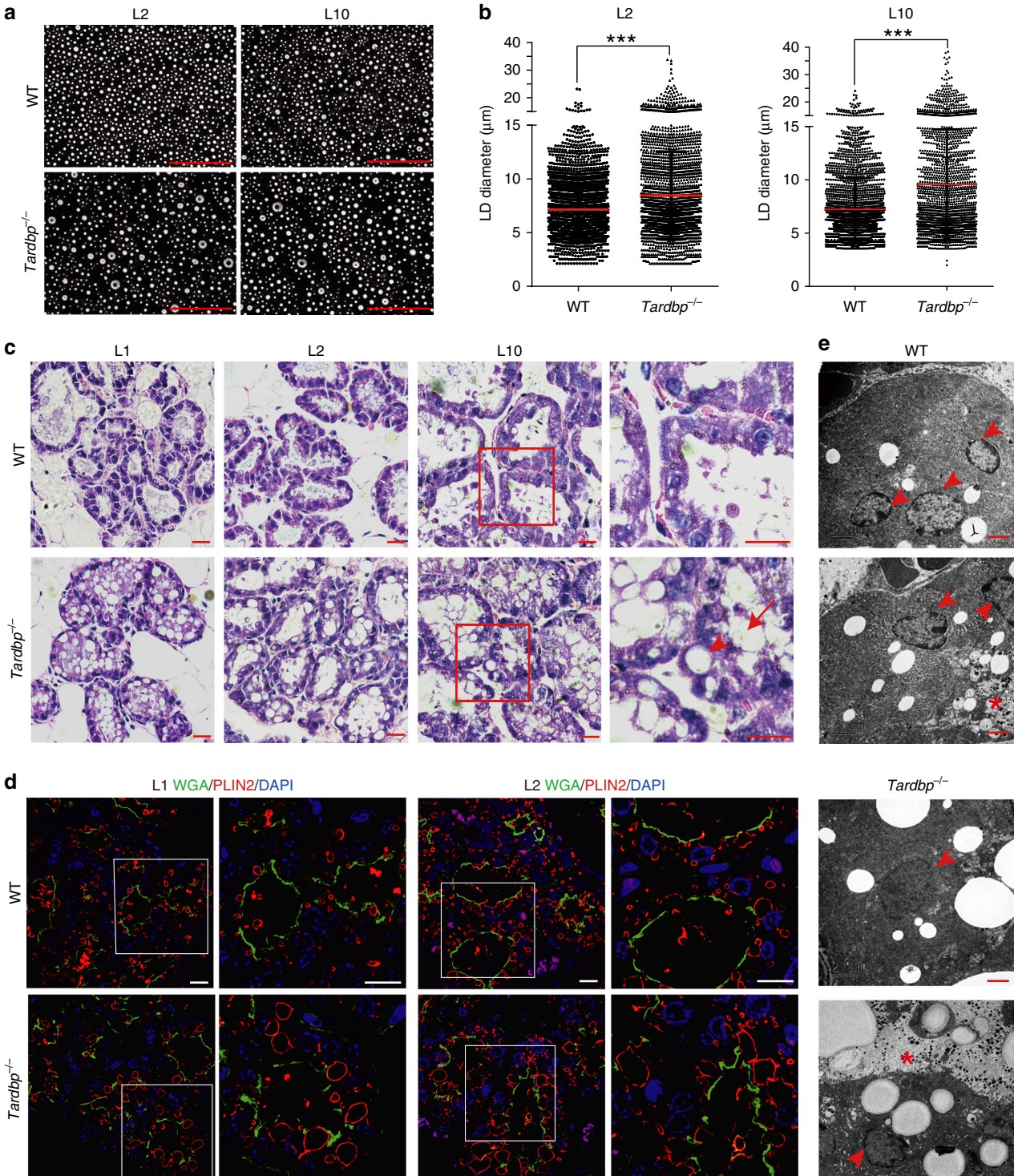

**Fig. 3 Larger lipid droplets are present in MECs after knockout of *Tardbp*. a, b** Phase contrast micrographs (**a**) and statistics (**b**) of milk fat globules (MFGs) from wild-type (WT) and *Tardbp⁻/⁻* mice analyzed using Image-Pro Plus 5.0 at lactation day 2 (L2) (left, *n* = 7 mice) or L10 (right, *n* = 6 mice). Scale bar: 100 μm. Data are means ± SD. Unpaired *t* test was used to evaluate statistical significance. \*\*\**P* < 0.001. **c** Hematoxylin and eosin staining of mammary glands from WT and *Tardbp⁻/⁻* mice at L1, L2, and L10. Magnified areas at L10 are shown in red boxes (left). Arrowhead shows cytoplasmic lipid droplets (CLDs) in cells and arrow shows large lipid droplets (LDs) in alveolar lumen. Scale bar: 20 μm. **d** Immunofluorescence staining of adipophilin (PLIN2) (red) showing LDs in alveoli of mammary glands from WT and *Tardbp⁻/⁻* mice at L1 and L2. Sections were also stained with wheat germ agglutinin (WGA) (green) and 4′,6-diamidino-2-phenylindole dihydrochloride (DAPI) to outline luminal border or nuclei. Magnified areas (right) are shown in white boxes (left). Scale bar: 10 μm. **e** Electron microscopy images of mammary gland sections from WT and *Tardbp⁻/⁻* mice at L1. Arrowhead indicates nucleus and asterisk shows lumen of mammary glands. Scale bar: 2 μm. Source data are provided as a Source Data file.

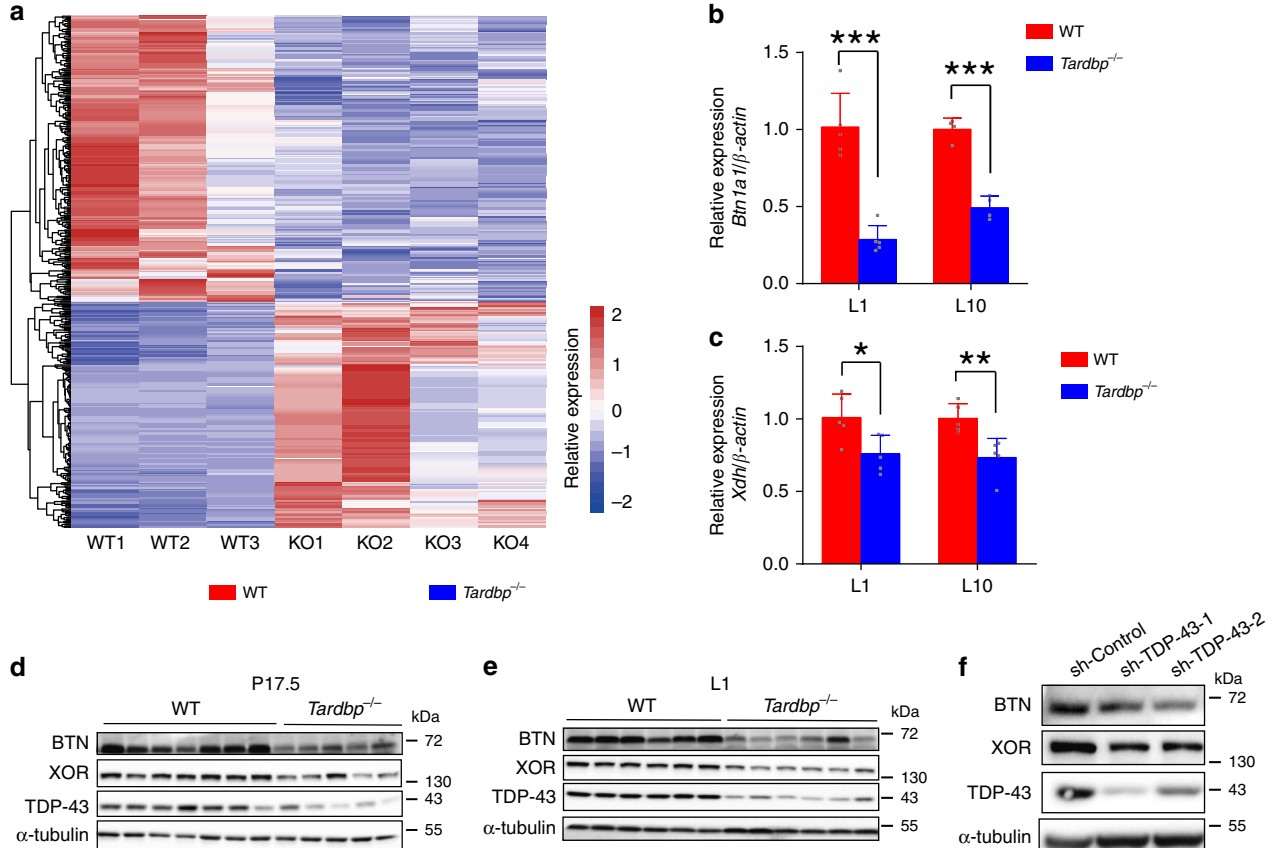

**Fig. 4 TDP-43 regulates expression of BTN and XOR. a** Heatmap of differentially expressed genes from mammary glands between wild-type (WT, $n = 3$) and $Tardbp^{-/-}$ (KO, $n = 4$) mice at lactation day 1 (L1). Colors of bars represent normalized and scaled expression levels (red, white, and blue correspond to highest, middle, and lowest values, respectively). **b** Quantitative real-time PCR (qRT-PCR) analysis of $Btn1a1$ expression in WT and $Tardbp^{-/-}$ mice at L1 ($n = 5$ mice) and L10 ($n = 4$ mice). **c** Quantitative real-time PCR analysis of $Xdh$ expression in WT and $Tardbp^{-/-}$ mice at L1 ($n = 5$ mice) and L10 ($n = 5$ mice). **d**, **e** Western blot showing expression of BTN and XOR in MECs isolated at pregnancy day 17.5 (**d**) and L1 (**e**) from WT and $Tardbp^{-/-}$ mice. **f** Protein levels of BTN and XOR in differentiated HC11 cell line upon $Tardbp$ knockdown. Data are means ± SD. Unpaired $t$ test was used to evaluate statistical significance. *$P < 0.05$; **$P < 0.01$. Source data are provided as a Source Data file.

lactation[32]. The LDs observed in the alveoli were indeed larger in size and accumulated around the edges of the lumen during lactation in $Tardbp^{-/-}$ mice (Fig. 3d and Supplementary Fig. 9B). Moreover, electron microscopy images of mammary gland sections from WT and $Tardbp^{-/-}$ mice further demonstrated that large LDs were accumulated in the MECs (Fig. 3e). Taken together, these data indicate that loss of TDP-43 resulted in the accumulation of large LDs in the MECs, which may eventually lead to lactation failure.

**TDP-43 loss reduces BTN and XOR expression.** To explore the mechanism of lipid secretion disorder in $Tardbp^{-/-}$ mice, RNA-sequencing (RNA-seq) was performed to identify the molecular processes at L1 between WT and $Tardbp^{-/-}$ mice. Among the differentially expressed genes identified in the $Tardbp$ KO epithelium (Fig. 4a), 44 genes were involved in lipid metabolism. It has been reported that the TDP-43 protein regulates downstream genes by directly binding to the motif of UG-enriched sequences[33]. Therefore, to identify genes directly regulated by TDP-43, we performed statistical analysis to determine enrichment of the UG-repeated motifs in the mRNAs of the above 44 genes (Supplementary Table 2). Results showed that $Btn1a1$ mRNA exhibited the most significant enrichment in the UG-repeated motif. Previous research has also shown that $Btn1a1$ KO mice exhibit poor pup survival and large lipid droplets[13], thus phenocopying

$Tardbp$ KO mice. This suggests that $Btn1a1$ may be a downstream gene of TDP-43 that regulates milk lipid secretion. In addition, qRT-PCR assay confirmed the significant decrease in $Btn1a1$ expression in the TDP-43 KO mammary gland at L1 and L10 (Fig. 4b). Moreover, $Xdh$, an important interactor with BTN for milk lipid secretion[34], was significantly decreased at L1 and L10 (Fig. 4c), whereas the expression of $Cidea$ was comparable at P17.5 and L10 at both the mRNA and protein levels (Supplementary Fig. 10A–C). We therefore focused on the $Btn1a1$ and $Xdh$ genes, which are well-recognized mediators of milk lipid secretion[32].

To confirm the regulatory effects on $Btn1a1$ and $Xdh$, we compared their expression levels between WT and $Tardbp^{-/-}$ mice. The mRNA levels of $Btn1a1$ and $Xdh$ decreased in the $Tardbp^{-/-}$ mammary gland during lactation; moreover, BTN ($Btn1a1$ protein) and XOR ($Xdh$ protein) expression were both remarkably reduced at P17.5 and L1 (Fig. 4d, e). To prove the regulatory effects in vitro, knockdown of $Tardbp$ by short hairpin RNA (shRNA) in a differentiated mammary epithelial cell line (HC11) also led to substantially lower expression of BTN and XOR at the protein level (Fig. 4f), thus suggesting a regulatory effect of TDP-43 on BTN and XOR. Knockdown of TDP-43 in the HC11 cells impaired dome formation (Supplementary Fig. 10D), which results from fluid secretion by mammary epithelial cells upon treatment with lactogenic hormones

in vitro[35–37]. Moreover, the decrease in dome formation was partially rescued when BTN and XOR were co-expressed in cells expressing sh-TDP-43 (Supplementary Fig. 10E, F).

**TDP-43 binds to *Btn1a1* and *Xdh* mRNA.** TDP-43 is a RBP[38], with the binding motif of the UG-enriched sequence[33,39]

regulating RNA in a variety of ways. To decipher the regulatory mechanism of TDP-43 on *Btn1a1* and *Xdh*, we examined whether a UG- or TG-enriched sequences existed in the *Btn1a1* and *Xdh* gene promoter regions, pre-mRNA, and mRNA. Both *Btn1a1* and *Xdh* mRNA contained a large TG-enriched sequence in the 3′-UTR (Fig. 5a), suggesting that TDP-43 may bind to the 3′-UTRs of the *Btn1a1* and *Xdh* transcripts directly. To demonstrate this,

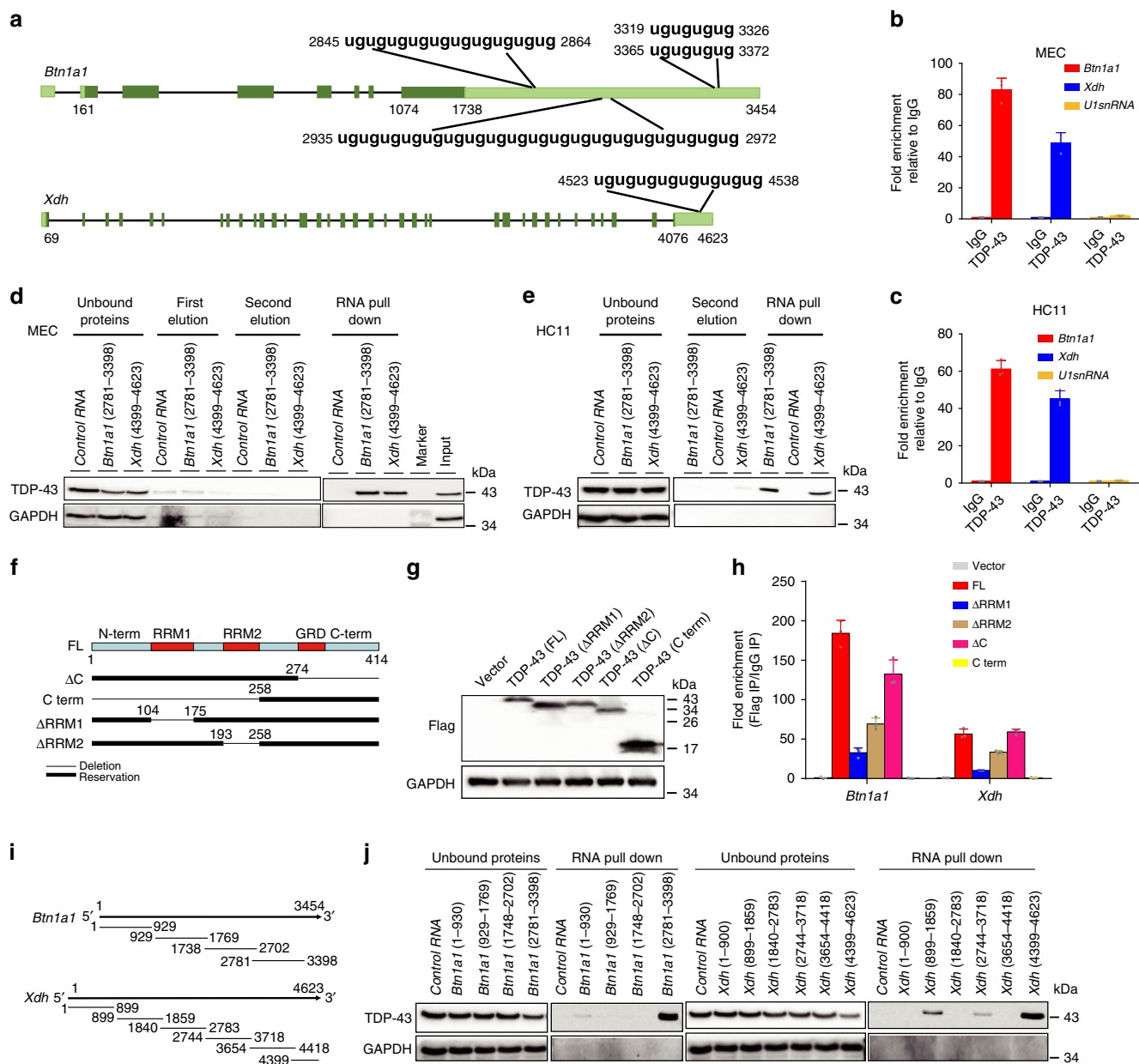

**Fig. 5 TDP-43 directly binds to *Btn1a1* and *Xdh* mRNA. a** mRNA structures of *Btn1a1* (upper) and *Xdh* (lower) showing TDP-43-binding motif in 3′-UTRs. Deep green boxes represent coding region. Light green boxes represent non-coding region. **b, c** RNA immunoprecipitation (RIP) assay for analysis of interaction between TDP-43 protein and *Btn1a1* or *Xdh* mRNA in MECs and differentiated HC11 cell line using TDP-43 antibody. **d, e** MECs (**d**) and differentiated HC11 (**e**) cellular extracts were incubated with in vitro-transcribed biotin-labeled control (pcDNA3.1 vector), *Btn1a1*, or *Xdh* mRNA fragments for biotin RNA pull-down followed by western blot analysis. Control RNA used biotinylated RNA without TDP-43-binding site (ugugug). **f** Structures of TDP-43 and their mutants used in RIP assays. TDP-43 protein contains an N-terminal domain (N-term), two RNA-recognition motifs (RRM1 and RRM2), a glycine-rich domain (GRD), and a C-terminal domain (C-term). **g** Western blot of overexpressed Flag-TDP-43 or its mutants in HC11 cell line. **h** RIP assay using Flag antibody in differentiated HC11 cells after overexpression of Flag-TDP-43 or Flag-mutants of TDP-43. **i** Schematic of mouse *Btn1a1* or *Xdh* mRNA fragments used for RNA pull-down assays. **j** Binding between TDP-43 protein and fragmented *Btn1a1* and *Xdh* mRNA. Four biotinylated *Btn1a1* mRNA fragments or six biotinylated *Xdh* mRNA fragments were incubated with HC11 cellular extract, and interaction between TDP-43 and each fragment was examined by RNA pull-down followed by western blot analysis. Data are shown as the means ± SD of three independent experiments. Source data are provided as a Source Data file.

we studied the interaction between the TDP-43 protein and mRNA of *Btn1a1* and *Xdh* by RNA immunoprecipitation (RIP). Results demonstrated that TDP-43 was bound to *Btn1a1* and *Xdh* mRNA strongly in the MECs (Fig. 5b) and HC11 cell line (Fig. 5c). An RNA pull-down assay was also used to confirm the interaction between mRNA and protein. Both *Btn1a1* (nt 2781–3398) and *Xdh* (nt 4399–4623) fragments containing TG-enriched sequences effectively pulled down the TDP-43 protein in the primary MECs and HC11 cells (Fig. 5d, e).

To determine which domains of TDP-43 were responsible for its interaction with *Btn1a1* and *Xdh* mRNA, Flag-TDP-43 or deletion fragments was overexpressed for the RIP assays using differentiated HC11 cell lysates (Fig. 5f, g). The C-terminal domain of TDP-43 (C-term) could not bind to either *Btn1a1* or *Xdh* mRNA, whereas the other deletion constructs showed clear binding affinity to both mRNAs. In addition, both the RRM1 and RRM2 domains were required for the interaction of TDP-43 with *Btn1a1* and *Xdh* mRNA. The RRM1 domain was the predominant functional RNA-binding domain, with RRM1 deletion (ΔRRM1) showing the greatest reduction in binding affinity compared with the TDP-43 full-length construct (FL) (Fig. 5h).

We further performed reciprocal RNA pull-down assays using four biotinylated *Btn1a1* mRNA fragments or six biotinylated *Xdh* mRNA fragments (Fig. 5i) to confirm the interaction between TDP-43 and *Btn1a1* and *Xdh* mRNA. Consistent with Fig. 5d, e, the 2781–3398 nucleotides of *Btn1a1* and 4399–4623 nucleotides of *Xdh* containing UG repeated at the 3′ end of the mRNA strongly interacted with TDP-43, although the 1–900 nucleotides of *Btn1a1* and 899–1859 and 2744–3718 nucleotides of *Xdh* also showed weak binding to TDP-43 (Fig. 5j).

Taken together, the above data demonstrated strong interactions between the TDP-43 protein and *Btn1a1* and *Xdh* mRNA.

**TDP-43 loss decreases *Btn1a1* and *Xdh* mRNA stability**. TDP-43 can regulate mRNA stability through the 3′-UTR of mRNA[40–42]. We therefore examined the decay of *Btn1a1* and *Xdh* mRNA following treatment with actinomycin D, a RNA Pol II inhibitor, to stop transcription in primary MECs from P17.5 and L1. Results showed that *Btn1a1* and *Xdh* mRNA declined significantly in the absence of TDP-43 between the WT group and *Tardbp* KO group at P17.5 and L1, and internal control *Gapdh* mRNA remained unchanged (Fig. 6a, b), suggesting that TDP-43 loss decreased the stability of *Btn1a1* and *Xdh* mRNA. Moreover, overexpressed TDP-43 full-length (Flag-FL) but not the C-terminal domain lacked the RRM1 and RRM2 regions, significantly suppressing the decay of *Btn1a1* and *Xdh* mRNA in the differentiated HC11 cell line (Fig. 6c). Both the BTN and XOR protein levels were remarkably increased upon TDP-43 overexpression compared with that in the control group (Supplementary Fig. 11A). These data indicate that TDP-43 could stabilize *Btn1a1* and *Xdh* mRNA levels.

To further demonstrate that TDP-43 regulates *Btn1a1* and *Xdh* mRNA stability through the 3′-UTR, we inserted the 3′-UTRs of *Btn1a1* or *Xdh* following the green fluorescent protein (GFP) gene into a mammalian expressive vector, representing GFP-Btn1a1-UTR and GFP-Xdh-UTR, respectively (Fig. 6d, upper), and then determined GFP change upon TDP-43 knockdown in the HC11 cells. As expected, knockdown of TDP-43 by two independent shRNAs led to substantially lower GFP expression in the GFP-Btn1a1-UTR and GFP-Xdh-UTR groups compared with their scramble controls (Fig. 6d), whereas GFP expression remained unchanged upon TDP-43 knockdown in the GFP-Control-UTR group. To confirm the regulation of TDP-43 on *Btn1a1* and *Xdh* mRNA stability, we next generated deletion

mutations of the *Btn1a1* and *Xdh* 3′-UTRs without TDP-43-binding sites (GFP-Btn1a1-mutUTR and GFP-Xdh-mutUTR, respectively). RIP-qPCR assays were then conducted to confirm that deletion mutations of the UG-enriched sequences could severely abate the interaction between GFP mRNA and the TDP-43 protein (Supplementary Fig. 11B). Mutation of the TDP-43 binding site completely abolished the down-regulation of GFP expression upon TDP-43 knockdown (Fig. 6d). To measure whether TDP-43 could regulate mRNA stability of the above GFP reporters, we detected the GFP mRNA expression of each group upon actinomycin D treatment after TDP-43 knockdown. Results showed that the GFP mRNA stability of GFP-Btn1a1-UTR and GFP-Xdh-UTR, but not GFP-Btn1a1-mutUTR or GFP-Xdh-mutUTR, significantly decreased upon sh-TDP-43 treatment compared with the control group (Supplementary Fig. 11C). To determine whether TDP-43 could regulate BTN and XOR expression in humans, we introduced human *BTN1A1* and *XDH* 3′-UTRs into GFP reporters to generate GFP-hBTN1A1-UTR and GFP-hXDH-UTR, respectively. Immunoblot analysis showed that knockdown of TDP-43 by independent shRNAs reduced GFP expression of GFP-hBTN1A1-UTR and GFP-hXDH-UTR relative to the sh-control (Supplementary Fig. 11D, E).

**TDP-43 loss results in early involution**. We also examined whether premature involution occurred during lactation in *Tardbp*−/− mice. The secretory lobuloalveolar structures were remodeled into glandular structures in *Tardbp*−/− mice, whereas the compact alveolar structures persisted in WT mice until L21. Loss of alveolar structures was also observed at L18 (Fig. 7a). Although *Tardbp*−/− mammary glands were similar to WT mammary glands in whole-mount staining at L15 (Fig. 7a), H&E staining at higher magnification showed that the *Tardbp*−/− mammary epithelium accumulated MFGs in the alveoli and exhibited cell shedding at L15 and L18 (Fig. 7b, c and Supplementary Fig. 12A). Moreover, severe tissue remodeling in the *Tardbp*−/− mammary epithelium eventually resulted in the collapse of many mammary alveoli at L21 (Fig. 7d). This tissue damage in *Tardbp*−/− mice was correlated with the time course of pup death (Fig. 1d). Moreover, we detected cell division using Ki67 staining during late lactation and found that cell proliferation did not change at L15 and L18, but decreased at L21 in KO mice (Supplementary Fig. 12B, C). The underlying mechanism requires further investigation.

Overall, these data demonstrated that depletion of *Tardbp* in the female mammary gland led to premature involution.

**Low TDP-43 expression relates to human lactation deficiency**. To explore the potential effect of TDP-43 expression on human milk secretion, we collected fresh milk samples from a total of 60 healthy lactating women who gave birth to a full-term infant. It has been reported that intracellular components of MECs, including mRNAs, can be trapped within MFGs during cellular MFG formation and secretion[43,44], and thus mRNA expression profiles from MFGs can be representative of MEC gene expression[45,46]. Therefore, to analyze the expression levels of *TARDBP* in human MECs during lactation, MFGs from human breast milk were obtained by centrifugation for RNA isolation. We then detected the MFG mRNA levels of cell-specific markers to rule out possible contamination of RNA from immune cells (Fig. 8a), which are abundant in human milk[47]. Questionnaire follow-up by telephone interview was performed to assess milk secretion in the mothers, which confirmed that partial breast-feeding and formula feeding were driven by necessity. Our results demonstrated that *TARDBP* was significantly up-regulated in the exclusive breastfeeding group in comparison to the partial

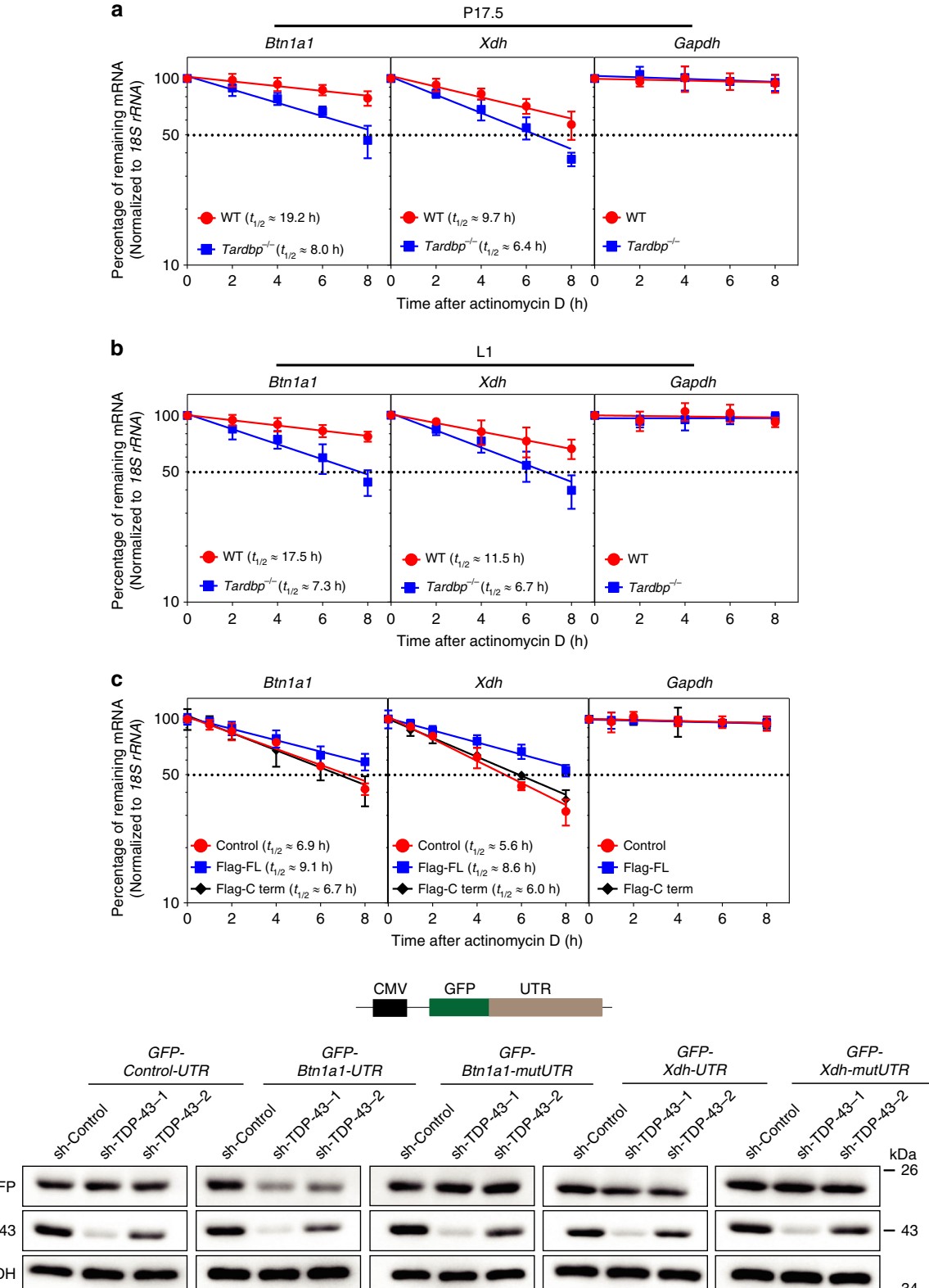

breastfeeding group (Fig. 8b). The expression levels of *HNRNPA1*, another RBP, showed no changes among the different breastfeeding groups (Fig. 8c). Furthermore, we found no significant differences among groups in terms of milk collection time and delivery type (Supplementary Fig. 12D, E). This result suggests that low expression of TDP-43 may be correlated with lactation deficiency in human milk secretion.

## Discussion

Milk secretion is critical for nutrition delivery from parent to offspring. Recently, researchers developed intravital imaging procedures using transgenic mice to provide mechanistic insight into the secretion of lipid droplets within the mammary epithelium in real time, which could be applied to investigate trafficking events during lactation[48,49]. However, the

**Fig. 6 TDP-43 loss decreases *Btn1a1* and *Xdh* mRNA stability. a, b** Primary MECs isolated at pregnancy day 17.5 (P17.5) (**a**) and lactation day 1 (L1) (**b**) from wild-type (WT) and *Tardbp*<sup>−/−</sup> mice were treated with actinomycin D for the indicated time, then *Btn1a1*, *Xdh*, and *Gapdh* mRNA expression levels were analyzed by qRT-PCR. Data were normalized to *18S rRNA* levels in each experiment and represented as a percentage of mRNA levels measured at time 0 h (before actinomycin D addition) using a semi-logarithmic scale. Half-lives ($t_{1/2}$) were calculated as time of each mRNA to decrease to 50% of its initial abundance. **c** qRT-PCR analysis of *Btn1a1* and *Xdh* mRNA expression in differentiated HC11 cells treated with actinomycin D after overexpression of control (Flag), Flag-TDP-43 full-length (Flag-FL), and Flag-TDP-43 C-term (Flag-C-term). **d** Upper: schematic of reporter constructs containing green fluorescent protein (GFP) gene fused with *Btn1a1* and *Xdh* mRNA 3'-UTRs. *GFP-Btn1a1-mutUTR* and *GFP-Xdh-mutUTR* were generated by deletion mutations of UG-enriched sequences in *Btn1a1* (position 2845–2864, 2935–2972, 3319–3326, and 3365–3372 nt) and *Xdh* (position 4523–4538 nt) 3'-UTRs, respectively. Lower: GFP expression levels were measured by western blotting 72 h after co-transfection with GFP reporter and sh-TDP-43. sh-TDP-43-1 and sh-TDP-43-2 represent independent shRNAs used. Data are shown as the means ± SD of three independent experiments. Source data are provided as a Source Data file.

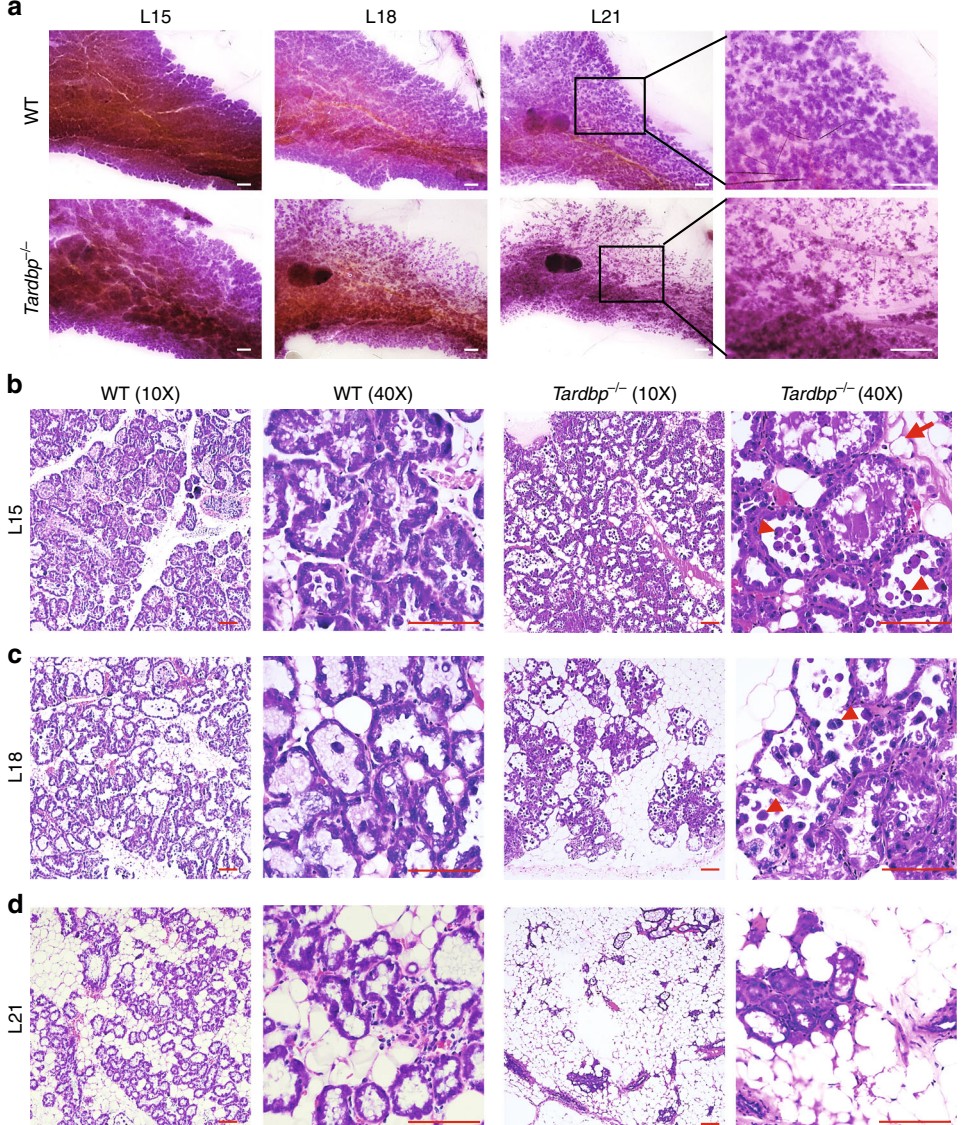

**Fig. 7 TDP-43 loss results in early involution. a** Whole-mount carmine staining of mammary glands of wild-type (WT) and *Tardbp*<sup>−/−</sup> mice at lactation day 15 (L15), L18, and L21. Magnified areas are shown in black boxes. Scale bar: 1 mm. **b–d** Paraffin sections of mammary glands taken at various times during lactation and stained with hematoxylin and eosin. Arrows show milk lipid droplets and arrowhead shows shedding cells. Scale bar: 100 μm.

fundamental regulation of milk secretion remains largely unknown. In the present study, we found an important participant (TDP-43) required for milk fat droplet secretion in the lactating mammary gland, which contributed to lipid secretion by regulating *Btn1a1* and *Xdh* mRNA stability. In addition, *Tardbp* KO resulted in a reduction in *Btn1a1* and *Xdh* mRNA stability and induction of lipid secretion failure in the mammary epithelium (Fig. 8d). These results highlight the the

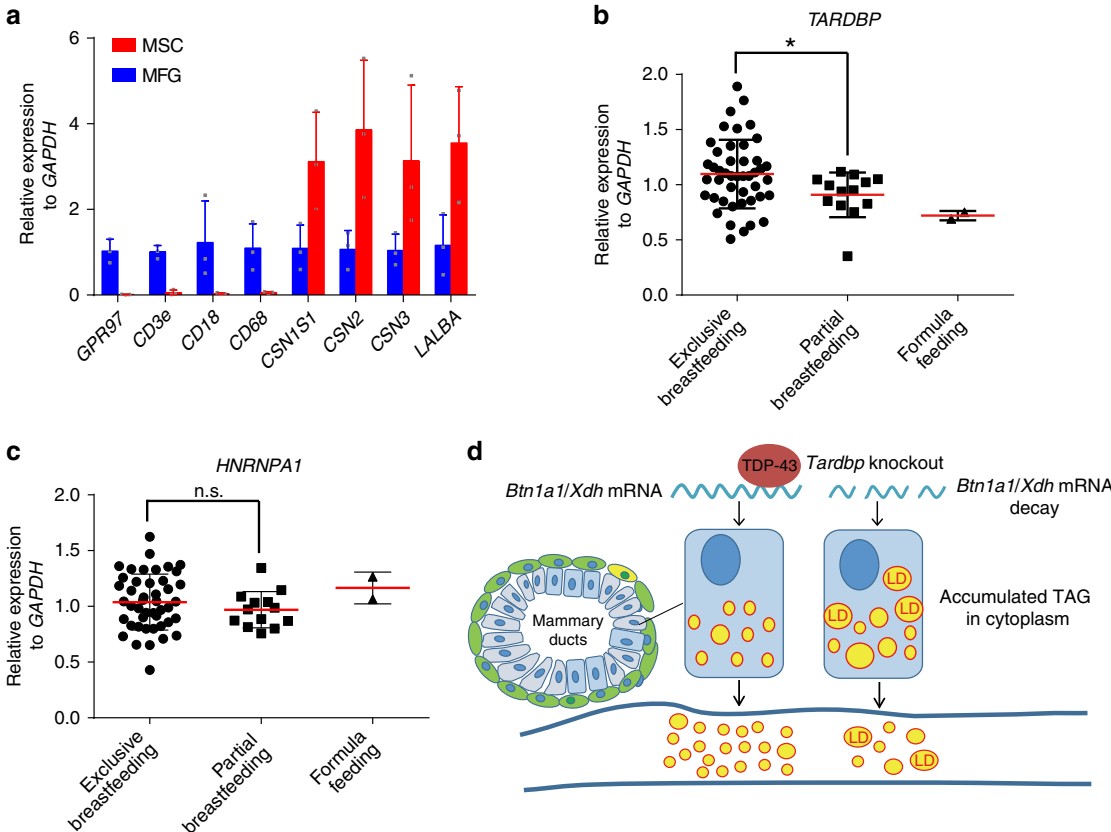

**Fig. 8 Low TDP-43 expression relates to human lactation deficiency. a** Quantification of mRNA expression levels of cell-specific markers in MSCs (milk somatic cells, $n = 3$) and MFGs (milk fat globules, $n = 3$). MSCs were obtained from fresh human breast milk by centrifugation. Specific markers for polymorphonuclear neutrophils (*CD18* and *GPR97*), lymphocytes (*CD3e*), macrophages (*CD18* and *CD68*), and MEC genes encoding milk proteins (*CSN1S1*, *CSN2*, *CSN3*, and *LALBA*) are shown. **b**, **c** qRT-PCR analysis of *TARDBP* (**b**) and *HNRNPA1* (**c**) mRNA expression levels in MFGs of fresh human milk on days 3–5 postpartum. Exclusive breastfeeding, $n = 45$; partial breastfeeding, $n = 13$; formula feeding, $n = 2$. **d** Graphic abstract of roles of TDP-43 in regulating milk lipid secretion. Data are means ± SD. Unpaired $t$ test was used to evaluate statistical significance. *$P < 0.05$; n.s., not significant. Source data are provided as a Source Data file.

critical role of TDP-43 in milk lipid secretion, thus enhancing our understanding of milk secretion.

Positive selection is a considerable evolutionary force behind the divergence of species[50,51], and genome-wide scans for positive selection of protein-coding genes are valuable tools for gaining insight into the genomic mechanisms of traits[52–54]. In the present study, we performed phylogenomic analysis of positive selection in 15 vertebrate genomes to identify candidate genes for milk lipid secretion. Our results revealed that *TARDBP* and *SRSF9* underwent significant positive selection. Furthermore, functional assays showed that the RBP TDP-43, but not SRSF9 (Supplementary Fig. 4A, B), regulated *Btn1a1* and *Xdh* expression and the process of milk lipid secretion.

The expression of BTN is vital for successful milk secretion, with previous studies on genetically engineered mice indicting that *Btn1a1* expression is essential for milk lipid secretion[13]. Furthermore, using mammary-specific XOR KO mice, Monks et al.[19] found that XOR modulates milk lipid secretion and lactation initiation, but deletion of *Xdh* in mammary epithelial cells does not prevent MFG secretion. To date, however, few studies have attempted to identify upstream regulatory factors of XOR and BTN. Encoded by the *Tardbp* gene, TDP-43 is well recognized as an essential factor in amyotrophic lateral sclerosis and frontotemporal lobar degeneration[55,56]. Previous studies have shown that TDP-43 is involved in the progression of breast and other cancers[29,57,58]. In our study, TDP-43 loss resulted in large lipid droplets, consistent with the phenotypes observed in

*Btn1a1*$^{-/-}$ mice[13]. TDP-43 positively regulated *Btn1a1* and *Xdh* expression by stabilizing their mRNA levels. Our findings emphasize the RBP TDP-43 as an important post-transcriptional regulator in the lactation process.

In the present study, many pups from *Tardbp* KO mothers died within the first 2 days of lactation (Fig. 1b, c), similar to the findings in *Btn1a1* KO mice[13]. This may be due to large lipid droplets accumulating in the mammary gland, resulting in glandular ducts becoming full of fat and pups being unable to obtain milk in the first 2 days. Once milk began to flow, the pup survival rate stabilized until L15. Thereafter, there was a noticeable drop in survival, especially when we adjusted the number of pups in each litter to seven at L2. These results suggest that accumulation of lipids within the mammary epithelium impaired milk secretion in *Tardbp* KO mice, and thus mothers could only maintain a certain number of pups (litter size) during late lactation.

Previous research has demonstrated that global heterozygous KO of *Xdh* results in premature involution during lactation[17]. However, in a mammary-specific XOR KO model, homozygous deletion of *Xdh* in the mammary epithelial cells results in only a modest lactation defect[19]. This discrepancy may result from the different KO models: the former study applied heterozygous deletion of XOR in the whole body, whereas the latter knocked out XOR specifically in mammary epithelial cells. These studies suggest that deletion of XOR in the mammary epithelial cells alone may not result in premature involution. In the present

study, both the pre-involution phenotype and down-regulation of XOR expression were observed in conditional TDP-43 KO mice; however, we cannot conclude that pre-involution of the *Tardbp* KO mammary gland was only mediated by regulation of *Xdh* expression. It is, therefore, possible that another mechanism also contributed to this phenotype.

Of note, TDP-43 is also involved in systemic lipid homeostasis. For example, in liver tissue, TDP-43 interacts with LncLSTR to regulate *Cyp8b1* expression, thereby altering bile acid composition and plasma triglyceride clearance in mice[59]. Mammary glands are highly dynamic tissues and the nutritional organ for fat metabolism, especially during lactation; however, whether TDP-43 exerts an effect on lipid metabolism in mammary gland development is unknown. In the present study, we found that loss of TDP-43 in the mammary gland led to lipid accumulation in MECs, which induced milk lipid secretion failure and poor newborn survival. These results suggest that TDP-43 may play a more wide-ranging role in the regulation of lipid homeostasis in biology than previously expected.

From our RNA-seq data, many genes involved in lipid synthesis were significantly decreased in *Tardbp*$^{-/-}$ MECs at lactation onset (L1), but not at the pregnancy stage (P17.5) (Supplementary Fig. 8D, E). This may be due to feedback by obstacles to lipid secretion. The transient accumulation of higher levels of lipids in the cytoplasm (Fig. 2c) produced a negative feedback for lipid synthesis, and thus reduced lipid synthesis-related gene levels at the onset of lactation. During the pregnancy stage, although *Tardbp* was knocked out by WAP-Cre, lipid secretion had not yet been activated, and thus did not produce a negative feedback for lipid synthesis genes.

In summary, we found that KO of RBP TDP-43 induced severely insufficient lipid secretion and eventually lactation failure in the female mammary gland. Mechanistically, we identified *Btn1a1* and *Xdh* as post-transcriptional downstream genes of TDP-43, which affected their mRNA stability. Our clinical data demonstrated that low TDP-43 expression may be associated with insufficient milk secretion. Furthermore, our results emphasized the important roles of TDP-43 in the post-transcriptional regulation of milk secretion, providing a potential strategy for screening clinical lactation insufficiency.

## Methods

**Evolutionary analysis.** A total of 15 vertebrate species were used for evolutionary analysis, including eight mammals (*Homo sapiens*, *Mus musculus*, *Canis lupus familiaris*, *Bos taurus*, *Dasypus novemcinctus*, *Loxodonta africana*, *Monodelphis domestica*, and *Ornithorhynchus anatinus*), covering most mammalian taxonomic groups and used as foreground branches, and four birds (*Gallus gallus*, *Meleagris gallopavo*, *Ficedula albicollis*, and *Taeniopygia guttata*), two reptiles (*Pelodiscus sinensis* and *Anolis carolinensis*), and one fish (*Latimeria chalumnae*) used as background branches. The topology of the 15 species from the Ensembl (http://asia.ensembl.org/index.html) species tree was used as an input tree in the PAML test[60]. A list of 80 human RBP genes[28] were used for evolutionary analysis to estimate the synonymous and nonsynonymous rates (dN/dS) and inference of positive selection by LRTs in the most recent common ancestor of mammals. After selecting one-to-one orthologous genes of human RBPs among the 15 species and the longest transcript of each in Ensembl, we removed the OGs, which contained less than nine orthologous genes, and which had no orthologous gene to the outgroup (*Latimeria chalumnae*). We used Prank[61] for multiple alignment of coding sequences (CDSs) and Gblock[62] to identify conserved codons with parameters of $-t = c$. The OGs with conserved alignment sequences <200 nt were removed. The dN/dS ratios ($\omega$) of the mammalian lineage for each gene were estimated by the branch-site model in PAML[60]. The log likelihoods of the positive detection model (lnL1: model A alternative hypothesis) and the corresponding null model (lnL0: model A null hypothesis) were calculated on the mammalian lineage of the 60 RBPs with the following control file in PAML[60]: Null hypothesis (branch-site model A, with $\omega2 = 1$ fixed) (model = 2, NSsites = 2, fix_omega = 1, omega = 1); alternative hypothesis (branch-site model A, with $\omega2$ estimated) (model = 2, NSsites = 2, fix_omega = 0, omega = 1). Here, $2\Delta$lnL is the double time of absolute lnL1 minus lnL0 and the *P* value is the right tail probability of the $\chi^2$ distribution of $2\Delta$lnL. The false discovery rate (FDR) was calculated[63] for

multi-correction for all orthologous genes of the RBPs. Genes with FDR <0.05 were identified as positively selected genes.

**Mice.** The *Tardbp* floxed mice were generated by the gene targeting approach using the BAC targeting vector to delete of exons 2 and 3 in the *Tardbp* mRNA as previously reported[30] and were then bred with WAP-Cre transgenic mice obtained from the Jackson Laboratory (008735). All experimental procedures and animal care and handling were performed per the protocols approved by the Ethics Committee of the Kunming Institute of Zoology, Chinese Academy of Sciences.

**Primary MEC preparation.** Mammary glands were minced and then digested in DMEM/F12 containing 5% fetal bovine serum (FBS), 1% penicillin–streptomycin–glutamine, 300 U/ml collagenase I (Sigma, C0130), and 100 U/ml hyaluronidase (Sigma, H3506) for 1–2 h at 37 °C. After digestion, a single-cell suspension was obtained by sequential incubation with 0.25% trypsin-EDTA for 3 min and 5 mg/ml dispase (Sigma, D4693) containing 0.1 mg/ml DNase I (Roche, 11248932001) for 5 min at 37 °C with gentle pipetting. Finally, red blood cells were removed in 0.8% NH$_4$Cl, followed by filtration through a 40-mm filter. Primary MECs were grown in DMEM/F12 medium with 10% FBS.

**Cell culture and mRNA stability analysis.** HC11 cell line was from Bernd Groner lab, Ludwig Institute for Cancer Research. The HC11 cells were grown in RPMI-1640 medium with 10% FBS, 5 µg/ml gentamycin sulfate, 5 µg/ml insulin (Sigma, I5500), and 10 ng/ml epidermal growth factor (EGF, Gibco, PGH0315). For differentiation induction, confluence cells were grown for 24 h in a medium without EGF supplementation, followed by growth in DIP medium (1 µM dexamethasone, 5 µg/ml insulin, and 5 µg/ml prolactin (ProSpec, cyt-240)) for the indicated time. To examine the decay of mRNA, cells were treated with 5 µg/ml actinomycin D and then collected at different times for RNA analysis.

**RNA extraction and PCR.** Total RNA was extracted using TRIzol reagent (Life Technologies) and then converted to complementary DNA using a PrimeScript™ RT Reagent Kit (TaKaRa, containing gDNA Eraser). We then performed qRT-PCR on a QuantStudio 3 instrument using a SYBR Green PCR Master Mix (Applied Biosystems). Primers used are listed in Supplementary Table 3.

**Whole-mount staining.** The fourth mammary glands were excised and spread on microscope slides and fixed in 25% glacial acetic acid and 75% ethanol for 1 h. Tissues were then stained in carmine alum solution overnight at 4 °C. After staining, the slides were dehydrated through increasing ethanol concentrations, cleared in xylene, and coverslipped with Neutral Balsam (Solarbio, G8590).

**Histology and immunostaining.** Mammary glands were fixed in formalin and embedded in paraffin. Tissue blocks were then sectioned at 5-µm thickness and stained with H&E. For immunostaining, sections were dehydrated with graded alcohol and boiled in 10 mM sodium citrate for antigen retrieval for 20 min. Sections were then used for immunohistochemical and immunofluorescence analyses. For immunohistochemistry, sections were incubated three times with 3% H$_2$O$_2$ for 5 min to inactivate endogenous peroxidases, and then blocked with 10% goat serum for 2 h and incubated with primary antibodies at 4 °C overnight. The slides were then washed three times in phosphate-buffered saline (PBS) and incubated with secondary antibodies for 1 h at room temperature and developed with 3,3′-diaminobenzidine. For immunofluorescence, sections were blocked for 2 h, and then incubated with primary antibodies for 2 h and secondary antibodies for 1 h at room temperature after direct antigen retrieval. The slides were counterstained with 4′,6-diamidino-2-phenylindole dihydrochloride (Vectashield, H-1200, Vector Laboratories). The primary antibodies used in immunostaining were TDP-43 (1:500) (Abcam, ab109535), K14 (1:1000) (Abcam, ab7800), K18 (1:1000) (Abcam, ab668), PLIN2 (1:1000) (Progen, GP40), milk antibody (1:500)(Nordic Immunology, 5941), Ki67 (1:1000) (Abcam, ab15580), and WGA (1:1000) (Life Technologies, 11261). The secondary antibodies used in immunostaining were fluorescein-labeled anti-rabbit (1:1000) (KPL, 02-15-06), fluorescein-labeled anti-mouse (1:1000) (KPL, 02-18-06), cy3 goat anti-mouse (1:2000) (Life Technologies, A10521), cy3 goat anti-rabbit (1:2000) (Life Technologies, A10520), and TRITC (tetramethylrhodamine isothiocyanate) rabbit anti-guinea pig (1:2000) (Life Technologies, A18888).

**Plasmid construction, knockdown, and overexpression.** For knockdown, small interfering RNAs (siRNAs) were purchased from RiboBio and shRNAs were cloned into the pLKO.1 vector. The siRNA or shRNA sequences are listed in Supplementary Table 3. For overexpression, FL and mutant mouse TDP-43 with Flag tags were cloned into pTRIPZ-inducible lentiviral expression vector, which can induce overexpression under 2 µg/ml doxycycline (Sigma, D9891). Sequencing verified all plasmid constructs to exclude mutations. These vectors were co-transfected with psPAX2 and pMD2.G (4:3:1) into 293 T cells to produce lentiviral particles, which were then transfected into HC11 cells. After 72–96 h, the cells were collected for further analysis or experiments. We used sh-GFP (Addgene #30323) and sh-TRC (Addgene #10879) as the shRNA controls and pTRIPZ-Flag-empty vector as the

overexpression control. For RNA pull-down assays, fragments of *Btn1a1* and *Xdh* were cloned into pcDNA3.1 under the T7 promoter. To generate the GFP reporter vector (pcDNA/GFP), the GFP fragment was cloned into pcDNA3.1(−) at the *Bam*HI and *Eco*RI sites. The pcDNA/GFP-Btn1a1-UTR construct was then created by inserting the Btn1a1 3′-UTR sequence (corresponding to NM_013483.3 from 2781 to 3398 nt) into the pcDNA/GFP vector at the *Eco*RV and *Sac*II sites. All primers used in this study are listed in Supplementary Table 3.

**Western blot analysis.** Protein lysates were electrophoresed by sodium dodecyl sulfate (SDS)-polyacrylamide gel electrophoresis and transferred to polyvinylidene difluoride membranes. Blots were incubated in 5% nonfat dry milk for 1 h and primary antibodies at 4 °C overnight, and then incubated with horseradish peroxidase (HRP)-linked secondary antibodies (Sigma) for 1 h at room temperature. Protein expression was detected using a chemiluminescent HRP substrate (Millipore). The antibodies used for immunoblotting were: Flag (1:2000) (CST, #14793), α-tubulin (1:5000) (Sigma, T5168), GAPDH (1:2000) (Santa Cruz, sc-25778), TDP-43 (1:2000) (Abcam, ab109535), XOR (1:2000) (Abcam, ab109235), and BTN (1:1000) (Acris, AP09532SU-N). Source data are provided as a Source Data file.

**RNA pull-down.** The mRNA fragments of *Btn1a1* and *Xdh* were synthesized in vitro using a HiScribe T7 High Yield RNA Synthesis Kit (NEB, E2040S). These RNAs were attached to a single biotinylated nucleotide at the 3′ terminus of the RNA strand using a Pierce RNA 3′ End Desthiobiotinylation Kit (Thermo, 20163). An RNA Pull-Down Kit (Thermo, 20164) was then used to detect the interaction between the TDP-43 protein and each fragment. Briefly, biotin-labeled RNAs were incubated with HC11 cellular extract for 1 h at 4 °C. The RNA–protein complexes were then washed three times with wash buffer, followed by protein elution and western blot analysis.

**RNA immunoprecipitation.** The RIP Kit (Millipore Cat. #17-701) was used for RNA immunoprecipitation assay according to the manufacturer's instructions. The MECs or differentiated HC11 cells were collected and lysed in RIP lysis buffer. The cell lysate was then immunoprecipitated with TDP-43 (1:10) (Proteintech, 10782-2-AP), Flag (1:50)(CST, #14793) (CST, #14793), or IgG (5 µg, Millipore, CS200621), and protein A/G magnetic beads for 6–8 h at 4 °C. The beads were washed with RIP buffer, followed by RNA purification and qRT-PCR analysis.

**Mouse milk collection and analysis.** Female mice were separated from their pups for 3–5 h at L2 or L10, anesthetized with xylazine at a dose of 10 mg/kg, and injected intraperitoneally with 10 or 0.2 U of oxytocin (Sigma) to induce milk let-down, as suggested by Wang et al.[14] and Boxer et al.[64]. After 10 min, milk was collected from the mammary glands. Milk collection was performed according to previously reported methods[65], with minor modification. One researcher held the anesthetized mouse while manually expressing the milk, and another researcher collected the milk using a P-200 Pipetman into a 1.5-ml tube. Each milk drop was transferred to the tube, which was then covered immediately to minimize liquid loss. Total time required to milk an animal was not longer than 40 min. All teats were milked in order starting with the right upper pelvic and going clockwise to the right upper axillary gland. After collection, the mice were dissected to observe whether all milk was completely collected, as assessed by eye and immuno-fluorescence staining using milk antibody. If milked incompletely, a large amount of milk could be seen within the mammary gland. Mice that retained milk within the mammary gland were excluded from further analysis. For milk protein analysis, protein concentration was determined using a Pierce BCA Protein Assay Kit (Thermo, 23228), with milk proteins separated on a 15% SDS-polyacrylamide gel and stained with Coomassie Brilliant Blue.

**TAG quantification and gas chromatography analysis.** The TAG content in intracellular MECs was determined using a Triglyceride Assay Kit (Applygen Technologies, E1013) according to the manufacturer's protocols. The TAG levels of each sample were normalized to the protein concentration measured by a BCA Protein Assay Kit. The fatty acid composition of milk was analyzed following published protocols[66], with minor modification. In brief, 20 µl of milk was added to 1 ml of MeOH (containing 2.5% $H_2SO_4$), then capped tightly, and heated for 60 min at 70 °C. The reaction mixture was then added to 0.2 ml of hexane and 1.5 ml of water and shaken vigorously. The solution was then centrifuged at $2500 \times g$ for 1 min at room temperature. The super phase containing fatty acid methyl esters (FAMEs) was removed into another tube, with 2 µl of FAMEs then injected for gas chromatography (Agilent 7890) analysis.

**Electron microscopy.** Procedures for electron microscopy were performed according to previous protocols[17], with minor modification. The fourth mammary glands were removed from WT and *Tardbp*$^{−/−}$ mice and fixed in 2% glutar-aldehyde and 1% paraformaldehyde in PBS. The samples were then post-fixed with 1% $OsO_4$ for 2 h at 4 °C, followed by serial ethanol dehydration and embedding in Epon 812 resin. Serial sections of uniform thicknesses (60 nm) were made using a Leica EM UC7 ultramicrotome. Ultrathin sections were then loaded onto 100-mesh Cu grids and double-stained with 2% uranyl acetate and lead citrate before

observations employing a JEM 1400 Plus transmission electron microscope at 120 kV.

**RNA-seq analysis.** Raw sequence data were processed through standard Illumina pipelines for base-calling and fastq file generation. Paired-end reads were mapped to the mouse genome primary assembly (NCBIM37) and the Ensembl mouse gene annotation for NCBIM37 genebuild was used to improve mapping accuracy with STAR v2.4.2a[67]. FeatureCounts v1.4.6-p5[68] was used to assign sequence reads to genes. Mitochondrial genes, ribosomal genes, and genes possessing less than five raw reads in half the samples were removed. Differential expression analysis was performed with the Bioconductor edgeR package v1.6[69]. Significant genes were determined by an adjusted *P* value of <0.01 based on Benjamini–Hochberg multiple testing correction and log-2-transformed fold change of >1 or <−1.

**Human milk sample analysis.** Fresh milk was obtained on days 3–5 postpartum from 60 healthy women who gave birth to full-term infants, and who exhibited similar time span of pregnancy and time of gestation. Written informed consent was obtained from all participants. In the early morning period, the donor manually pumped 3–5 ml of breast milk into a sterile, RNase-free collection tube, which was immediately placed on ice. Subsequent procedures of MFG isolation were finished within 3 h, with samples then transferred to a −80 °C freezer. We employed the RNA sampling method proposed by Maningat et al.[11]. Milk was mixed well before centrifugation at $1000 \times g$ for 10 min at 4 °C. The lipid fraction containing MFGs in the upper phase was transferred to another 2-ml RNase-free tube. The MFGs were washed with cold PBS, and then 1.5 ml of TRIzol lysis reagent was added to the lipid fraction for RNA extraction according to the manufacturer's instructions. RNA quality was checked by 2% agarose gel electrophoresis. A questionnaire follow-up by telephone interview was performed during the 7–8-week lactation period, which confirmed that partial breastfeeding and formula-feeding mothers were driven by necessity. The Ethics Committee of the Third Affiliated Hospital of Chongqing Medical University approved the project.

**Reporting summary.** Further information on research design is available in the Nature Research Reporting Summary linked to this article.

## Data availability

The RNA-seq data were deposited in the NCBI GEO database under ID code: GSE116456. The data that support our findings in this study are available from the corresponding author on reasonable request. The source data underlying Figs. 1b–f, 2a–d, 3b, 4b–f, 5b–e, g, h, j, 6a–d, 8a–c and Supplementary Figs. 4A, B, D, F, 5C–E, G–J, 6E, F, 7A–E, 8A, D, E, 10A–C, E, F, 11A–E, 12A, C–E are provided as a Source Data file.

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

## Acknowledgements

This work was equally supported by the National Key Research and Development Program of China (Grant No. 2016YFA0100900), Strategic Priority Research Program of the Chinese Academy of Sciences (Grant No. XDB13030400), National Science Foundation of China (Grant Nos. 31970612, U1802285, 81902714, 31801249), Yunnan Fundamental Research Projects (Grant Nos. 2018FA002, 2015HA026), Open Project from the State Key Laboratory of Genetic Resources and Evolution (Grant No. GREKF16-13 to L.A.). We thank Xun Huang, Bin Liang, Peng Shi, Ceshi Chen, and Wen Wang for constructive suggestions and Christine Watts for English editing. We would like to thank the Kunming Biological Diversity Regional Center of Instruments, Kunming Institute of Zoology, Chinese Academy of Sciences for our Electron Microscopy work and we would be grateful to Yingqi Guo for her help of making EM sample and taking images. We would like to thank Jieyu Wu and Xue Jiang for help with gas chromatography analysis.

## Author contributions

L.Z., H.K. and B.J. designed the experiments, interpreted the results, and wrote the manuscript. L.Z. and H.K. performed the experiments. G.-D.W. and L.W. performed the evolutional analysis. P.Y., S.X., M.L., L.P. and M.Z. collected clinical milk samples. H.X. and H.Z. analyzed the RNA-seq data. L.A., L.L., Q.Y. and L.Z. provided experimental assistance. C.-K.J.S. provided the *Tardbp* conditional floxed mice and discussed the results.

## Competing interests

The authors declare no competing interests.
