## [Peer Review File · Nature Communications]

Reviewers' comments:

Reviewer #1 (Remarks to the Author):

This paper presents potentially significant data on the function of the RNA-binding protein, TDP-43, in regulating the stability of mRNAs encoding butyrophilin 1a1 (Btn1a1) and xanthine oxidoreductase (Xdh). Btn1a1 and Xdh are proposed key players in the regulation of lipid secretion from mammary epithelial cells during lactation but nothing is known about how the amounts of these proteins are controlled at either the transcriptional, post-transcriptional or post-translational levels in milk-secreting mammary cells. Thus, the possibility that mRNA stability is an important factor is noteworthy. Furthermore, lactation is a crucial physiology for mammalian survival and thus of general scientific interest (and incidentally the basis for the worldwide dairy industry). Despite its biological importance, the cell biology of milk secretion has been neglected in recent years, so novel research in this area is welcome. However, there are many problems with this manuscript in its current form, especially with respect to some of the approaches taken and interpretation of the data. In addition, the English needs severe editing to be brought up to an acceptable standard.

Problems with some approaches and interpretation of data:

(1) The efficiency of Tardbp excision by Cre is shown qualitatively in Fig. S2 A, B. An estimate of % ablated cells would be useful as this is fundamental to the magnitude of phenotypic differences observed between wild type and mutant mice.

(2) An important reference has been missed in which Xdh was specifically ablated in the mammary gland [Monks, J. et al. (2016) *J. Physiol.* 594, 5899-5921]. The results are significantly different from the original description of the Xdh +/- mouse described by Vorbach, C. et al. [(2002) *Genes Dev.*, 16, 3223-3235] and more relevant as the lactation phenotype of the knock-out was directly compared with wild type animals (rather than comparing the heterozygote versus wild type in the original global knock out strain, the homozygote of which did not survive to breeding age).

(3) Page 4, lines 13-16. In Fig. 1A, histochemical staining of epithelial cells in the virgin tissue is quite pronounced and qualitatively does not seem that different from the pregnant sample, despite claims to the contrary. Furthermore, the author's claims are not confirmed by the supplementary data in Fig S1A and S1B. There is less mRNA for TDP-43 during lactation than in the virgin (additionally, statistical significance and animal numbers are not stated, neither is the lactation stage) and in the Western blot there is marginally less TDP-43 on lactation day 10 than in the virgin (it is questionable that tubulin is an appropriate loading marker for mammary tissue undergoing such marked changes in differentiation status, from the virgin to involuting state). The Western data should be quantified with an appropriate number of replicates and loading control.

(4) Page 5, lines 7-9 (Fig. 1 B-D). When comparing wild type and knock-out lactating animals it is standard procedure to adjust the litter sizes to the same numbers as soon as possible post-partum. So the data in Fig. 1B and C when this was not done are compromised. What appears to be happening is rapid death of the pups (whole litters, or a fraction of each litter?) because the glandular ducts are full of fat and the pups never get any milk out on the first two days. If milk begins to flow then many pups survive to weaning, albeit, underweight [this phenomenon was seen with the Btn1a1-/- mouse, Ogg, S.L. et al. (2004) *PNAS*, 101, 10084-10089]. In disagreement with the authors statement that "the surviving pups continue to perish from severe malnourishment until L15", the data show 100% survival between days 2-15, followed by a noticeable decline thereafter. The data in Fig. 1D when the litters were adjusted on day 2 are significantly different. In fact there is a 100% survival rate from day 1 until day 13. How do the authors explain this? This experiment would best be repeated with adjusted litter sizes and additional animals. A tangential but related issue:- did the pups have accessibility to solid food? This can be a confounding factor towards the end of lactation when pups may "self-wean" if solid

food is available.

(5) Page 6, lines 13-15. It would be more reassuring if the data on apoptosis and cell division were quantified (e.g., by counting TUNEL and Ki67 positive nuclei) with sufficient replicates for statistical significance. This also applies in late lactation (Page 12, lines 18-21 and Page 13, lines 1-3, Fig. 7). In this latter case, compensatory cell division may also be occurring, which could be confirmed using Ki67 as a marker [unpublished data of this reviewer, shows both increased apoptosis and cell renewal occurring in the *Btn1a1*^{-/-} mouse during peak lactation].

(6) Page 6, lines 18-19, and Page 22, lines 18-22 (Fig. 2). How was milk quantitatively recovered from mice, especially at day 2 of lactation? This is technically very difficult. Was a vacuum pump used? If so, how were evaporative losses controlled? Pup weight can serve as a useful indirect way of comparing milk yields among test and control groups, so it is surprising that there is no correlation between the milk yield differences on lactation day 2 in Fig. 2 with the pup weight data in Fig. 1. Another potential problem is the amount of oxytocin used to stimulate milk let-down if the doses used were 10 international units. If so, this is at least 50-fold more than required and can precipitate exfoliation of the epithelium (especially important if these mice were used for further experiments after lactation day 2 – e.g., milking on day 10).

(7) Page 8 Lines 3-9 (Fig. 3). Lipid droplet size differences appear comparable (although somewhat less) with the *Btn1a1*^{-/-} mouse [Ogg, S.L. et al., *ibid*], which deserves some comment as you would expect the two mutant lines to phenocopy each other since the mRNA and protein levels of *Btn1a1* are depleted in *Tardbp*^{-/-} mice (Fig. 4B,E).

(8) Page 9. Lines 1-3 (Fig.4). The RNAseq data are not adequately discussed. What other differences were noted related to lipid synthesis and secretion? Apparently quite a few, judging from later comments on page 15, line 15 - page 16, lines 1-3. What objective criteria were used to select *Btn1a1* and *Xdh* at the expense of other potential candidates?

(9) Page 12, lines 4-6 (Fig. 8D). Reference to this figure is out of place here and belongs in the Discussion with a broader discussion of mRNA regulation by TDP-43 in other contexts.

(10) Page 12, lines 9-11. This is one of several places where the Monks paper (see comment 2, above) should be highlighted with appropriate changes to the text.

(11) Page 13, lines 17-22, page 14, lines 1-3 (Fig. 8). It is not clear how the human subjects were followed up 7-8 weeks post-partum. Were these women formula feeding by choice or necessity? If by choice, then the data are meaningless. If by necessity, the data may have some merit, although there are only two 100% formula feeders, which cannot approach statistical significance.

(12) Page 15, lines 9-11. The severe premature involution observed in Vorbach's *Xdh*^{+/-} mouse does not agree with the more recent conditional *Xdh*^{-/-} line described by Monks et al.

(13) Page 15, line 15. A more complete discussion of the functions of TDP-43 in lipid homeostasis might help put the current data in a wider context.

(14) Page 16, lines 13-19. Unless the data on progesterone are shown in Results, this section of the Discussion needs to be deleted. Also, progesterone levels go down at parturition, at the time that TDP-43 should be increasing, so how can it control TDP-43 during lactation?

(15) Fig. S4A. Lactation day 2?

(16) Fig. S5. Gene expression and immunofluorescence data are not proxies for the measurement of protein synthesis or secretion.

Reviewer #2 (Remarks to the Author):

In this work, Zhao et al. have investigated the role of mouse TDP-43 in facilitating milk lipid secretion by post-transcriptional regulation of two genes, Xdh and Btn1a1 that are required for the secretion of lipid droplets from epithelial cells into the lumen. Following evolutionary analysis, the authors suggest that for this function TDP-43 has undergone positive selection in mammals and that knockout of the Tardbp gene in mice resulted in lipid droplet secretion failure associated with poor newborn survival. Finally, clinical samples from lactating women further substantiate the roles of TDP-43 in milk secretion.

Overall, the results of this study could be interesting but the mRNA aspects are not fully convincing:

1) In the evolutionary analysis, the authors identify both TARDBP and SRSF9 as undergoing positive selection. However, the possible role of SRSF9 is not further investigated, although it is an RNA binding protein like TDP-43 and could be affecting processing of the Btn1a1 and Xdh genes as well. Has this been investigated at all by the authors using a simple approach such as knocking down this factor (either alone or together with TDP-43) in HC11 cells and see whether these two genes might have a synergistic effect?. On page 4 the authors justify looking at TDP-43 because of previous work that suggest a role of this protein in breast cancer progression. However, also SRSF9 has been found to be altered in many kind of cancers and therefore in this respect there does not seem to be a big difference between the two.

2) In Figure 6, the authors show that TDP-43 can bind the 3'UTR of Btn1a1 and Xdh and affect the mRNA stability of the transcripts. However, the results of the experiments are not fully convincing. For example, using Flag-TDP-43 the authors show that the relative expression of Btn1a1 and Xdh can be increased following overexpression of TDP-43. However, because the authors just report a ratio, this experiment does not really show that TDP-43 can successfully recover the expression of wild-type protein levels, only that there is a rather moderate increase in expression. Has this experiment been performed?. Ideally, the authors should transfect HC11 cells with TDP-43 and show that Btn1a1 and Xdh protein expression can be fully recovered to wild-type levels. As a sidenote, using Flag is also not the ideal control for these experiments. As the authors hypothesize that TDP-43 mRNA binding is instrumental in affecting mRNA stability a much more convincing control would be a Flagged TDP-43 that carries mutations in the RRM1 and RRM2 regions.

3) In the 293T cells transfected with the GFP constructs carrying the 3' UTR of the Btn1a1 mRNAs, Figure 6, a critical control that is lacking and is represented by a control where the UG-rich regions in this 3'UTR have been deleted. One would then expect that the regulatory property of TDP-43 would be lost. Secondly, why was this reporter experiment also not repeated for the 3'UTR of Xdh?.

4) Finally, are the UG-repeated sequences in the human Btn1a1 and Xdh 3'UTRs conserved with respect to the mouse genes?. If not, then the connection between the mouse data and the milk yield in post-partum women shown in Figure 7 may not be very compelling. Even if they are conserved, the authors should show that the human genes are also positively regulated by human TDP-43 in the same way as the mouse genes

Reviewer #3 (Remarks to the Author):

The authors have identified the RNA-binding protein TDP-43 as a major regulator of milk production in mice and provide evidence that Btn1a1 and Xdh 1 mRNAs are effectors of this phenotype.

The overall findings of the report are novel and interesting. While the molecular players were previously known to participate in mRNA regulation (TDP-43), lipid homeostasis (BTN1A1), the implication of XDH1 in milk production is new. The work generally supports the model proposed by the authors, but several molecular specifics need to be strengthened and completed.

1. In Figures 5B, 5C, and 5H, the customary presentation of RIP data is as 'relative enrichment' of a given mRNA. The levels of that mRNA in IgG IP are set as '1' and the relative levels of the same mRNA in TDP-43 IP are shown as 'fold' difference relative to '1'.

2. Throughout the figures the authors should use the same nomenclature they use in the text: lowercase italics *Btn1a1* and *Xdh1* when referring to mouse gene, mRNA and fragments of mRNA. This nomenclature is not used correctly in the figures, although it is overall used correctly in the text.

3. In Figure 6, the representation of stability data also needs revision. These data should include decay rates for *Gapdh*, *Btn1a1* and *Xdh1* mRNAs with the following modifications:

- Control groups (before silencing) should be plotted for each mRNA in order to know their intrinsic stability before silencing TDP-43.
- Silencing groups should be plotted for each mRNA in order to assess their stability after silencing TDP-43. Typically, for a given mRNA the control siRNA group and the specific siRNA group are plotted in the same graph.
- Plotting should be done on a semilog scale (log Y axis, linear X axis).
- Half-lives should be determined for each mRNA as they cross the 50% line ($t_{1/2}$)

4. In Figure 6, the authors must complete the analysis of stability determinants in *Btn1a1* and *Xdh1* 3'UTRs. They need to prepare and test reporter constructs in which the full-length 3'UTR for each mRNA is assayed as well as reporters in which the major RNA segments in the 3'UTRs found to interact with TDP-43 (as found in Figure 5) are mutated. They then should assay if wild-type and mutant 3'UTRs display changes in (1) TDP-43 binding and (2) the stability of each reporter mRNA.

5. In Figure 7, can the milk production phenotype rescued in the *Tardbp*^{-/-} mice if *Btn1a1* and *Xdh1* mRNAs are re-expressed? For example, the authors could deliver via lentivirus *Btn1a1* and *Xdh1* mRNAs that lack the 3'UTR and hence are intrinsically stable. This experiment would provide critical support to the authors' model.

6. Although the text reads well, some attention to grammar, syntax, and word choice is needed. One example is the authors' use of the word 'imply', when they mean instead 'suggest' or 'indicate'.

Reviewer #4 (Remarks to the Author):

The authors analyzed the role of RNA-binding protein (RBP) TDP-43 in milk lipid secretion in mammals. Most of the manuscript concerns functional and molecular analyses of this RBP and its encoding gene, in particular using a KO mouse model. These analyses are far from my expertise and will be carefully addressed by other reviewers. I concentrate my analysis of the manuscript on its (very) short first part (pp 3-4), which deals with evolutionary analysis of RBPs in mammals. This analysis led the authors to conclude that TDP-43 underwent positive selection in mammals and this conclusion is presented as a main reason to further study the role of this particular RBP in milk lipid production.

There are two problems with this section of the manuscript.

First is about the way this analysis is presented. I don't think that the authors can claim, based on their very limited analysis, that « we performed the evolutionary analysis to study the molecular mechanism of milk secretion » (page 3 lines 6-7). Not sure neither that they can claim this analysis allowed to identify « the essential post-transcriptional regulator in mammals » (page 3 line 18). I think that the authors should explain in a more convincing way why this analysis is useful and why genes undergoing positive selection could be good candidate genes involved in lactation in mammals.

Second problem is that from what is shown in the manuscript and supplementary data, it is not very clear what has been done to select the gene families for which dN/dS has been analyzed. It should be explained what is the « stringent filtering » (page 3 line 20) that has been used. It is not clearly explained what is reported in the supplementary file in columns « Model A alternative » and « Model A null » and the calculated dN/dS values are not shown anywhere. At least for the two gene families on which emphasis is put (TARDBP and SRSF9), multiple alignments should be provided as supplementary figures.

Reviewers' comments:

Reviewer #1 (Remarks to the Author):

This paper presents potentially significant data on the function of the RNA-binding protein, TDP-43, in regulating the stability of mRNAs encoding butyrophilin 1a1 (Btn1a1) and xanthine oxidoreductase (Xdh). Btn1a1 and Xdh are proposed key players in the regulation of lipid secretion from mammary epithelial cells during lactation but nothing is known about how the amounts of these proteins are controlled at either the transcriptional, post-transcriptional or post-translational levels in milk-secreting mammary cells. Thus, the possibility that mRNA stability is an important factor is noteworthy. Furthermore, lactation is a crucial physiology for mammalian survival and thus of general scientific interest (and incidentally the basis for the worldwide dairy industry). Despite its biological importance, the cell biology of milk secretion has been neglected in recent years, so novel research in this area is welcome. However, there are many problems with this manuscript in its current form, especially with respect to some of the approaches taken and interpretation of the data. In addition, the English needs severe editing to be brought up to an acceptable standard.

Response: Thank you for the positive comments and highlighting areas for improvement. In the new manuscript, we have optimized experimental approaches and revised interpretation of the data, as discussed in the following paragraphs. In addition, the whole manuscript has been edited by a native English speaker to improve the overall language.

(1) The efficiency of Tardbp excision by Cre is shown qualitatively in Fig. S2 A, B. An estimate of % ablated cells would be useful as this is fundamental to the magnitude of phenotypic differences observed between wild type and mutant mice.

Response: Thank you for this suggestion. The statistical analysis of knockout (KO) efficiency was performed for five samples per group, as shown in Figure S5C in the revised manuscript.

(2) An important reference has been missed in which Xdh was specifically ablated in

the mammary gland [Monks, J. et al. (2016) J. Physiol. 594, 5899-5921]. The results are significantly different from the original description of the Xdh +/- mouse described by Vorbach, C. et al. [(2002) Genes Dev., 16, 3223-3235] and more relevant as the lactation phenotype of the knock-out was directly compared with wild type animals (rather than comparing the heterozygote versus wild type in the original global knock out strain, the homozygote of which did not survive to breeding age).

Response: Thank you for the references mentioned. We have cited and further discussed these papers in our revised manuscript (Pages 2, 17, and 18).

(3) Page 4, lines 13-16. In Fig. 1A, histochemical staining of epithelial cells in the virgin tissue is quite pronounced and qualitatively does not seem that different from the pregnant sample, despite claims to the contrary. Furthermore, the author's claims are not confirmed by the supplementary data in Fig S1A and S1B. There is less mRNA for TDP-43 during lactation than in the virgin (additionally, statistical significance and animal numbers are not stated, neither is the lactation stage) and in the Western blot there is marginally less TDP-43 on lactation day 10 than in the virgin (it is questionable that tubulin is an appropriate loading marker for mammary tissue undergoing such marked changes in differentiation status, from the virgin to involuting state). The Western data should be quantified with an appropriate number of replicates and loading control.

Response: Thank you for your questions and comments. In the initial Figure 1A, it is true that immunohistochemical (IHC) staining of TDP-43 was not particularly distinguishable. The brownness (color of substrate DAB for IHC) in the virgin stage (V, dark brown) was less strong than that in the pregnant stage (P17.5, pregnant day 17.5, nearly black), which was due to the over-development of DAB staining. To clarify, we repeated the experiment, with the results shown in Figure 1A (40× objective magnification) and Figure S4A (10× objective magnification) in the revised manuscript. The new data clearly demonstrated that TDP-43 expression levels in the virgin stage (V) were significantly lower than those in pregnancy (P17.5) and early lactation (lactation day 1 (L1) and 2 (L2)), but equivalent to that on L10. This pattern was also

confirmed by mRNA level detection (qPCR, Figure S4B).

We apologize for the unclear presentation of data in the initial Figure S1A. “Lactation” refers to “Lactation day 10 (L10)”, and all stages had three replicates. All data have been clearly labelled in the revised manuscript, as shown in Figure S4B.

We agree with your suggestion that Western blotting data should be quantified with an appropriate number of replicates and loading control. As such, we loaded more samples at every stage (V, P17.5, L1, L2, L10, and I3, four to five replicates for each stages) for TDP-43 antibody blotting (Figure S4C). A new internal control (β -actin) together with α -tubulin was applied to validate the loading amounts. The same amount of protein (40 μ g/lane) was loaded in each lane of the Western blots. The slightly decreased α -tubulin signals at late pregnancy and lactation may have resulted from dilution by the abundant expression of milk protein at these developmental stages. We also quantified the Western blotting results (Figure S4D), and all protein levels detected here demonstrated similar patterns to those in the qRT-PCR and IHC experiments.

*(4) Page 5, lines 7-9 (Fig. 1 B-D). When comparing wild type and knock-out lactating animals it is standard procedure to adjust the litter sizes to the same numbers as soon as possible post-partum. So the data in Fig. 1B and C when this was not done are compromised. What appears to be happening is rapid death of the pups (whole litters, or a fraction of each litter?) because the glandular ducts are full of fat and the pups never get any milk out on the first two days. If milk begins to flow then many pups survive to weaning, albeit, underweight [this phenomenon was seen with the *Btn1a1*^{-/-} mouse, Ogg, S.L. et al. (2004) PNAS, 101, 10084-10089]. In disagreement with the authors statement that “the surviving pups continue to perish from severe malnourishment until L15”, the data show 100% survival between days 2-15, followed by a noticeable decline thereafter. The data in Fig. 1D when the litters were adjusted on day 2 are significantly different. In fact there is a 100% survival rate from day 1 until day 13. How do the authors explain this? This experiment would best be repeated with adjusted litter sizes and additional animals. A tangential but related issue:- did the pups have accessibility to solid food? This can be a confounding factor towards the end*

of lactation when pups may “self-wean” if solid food is available.

Response: Thank you for your suggestions and critical questions regarding survival data.

In the original statement “the surviving pups continue to perish from severe malnourishment until L15”, we apologize for the incorrect description. At the end of L2, we adjusted the litter size to seven to observe the weights of the surviving pups (initial Figure 1E). Most pups died before L2 (which may be due to your explanation that the glandular ducts were full of fat and the pups did not receive any milk in the first two days, but survived once milk began to flow), but for those survived pups, we did NOT observe any pup deaths (100% survival rate) between L2-L13 (initial Figure 1D). However, the survival rate decreased again from L15 to L23 (initial Figure 1D). Therefore, the initial description “the surviving pups continued to perish from severe malnourishment until L15” should be “those pups that survived past L2 began to die from severe malnourishment at L15”. We apologize for the confusion. Moreover, this decrease in the litter survival rate at late lactation (L15-L21) only appeared for litter sizes with $n = 7$ and 8 , but not for smaller litter sizes ($n = 6$) (Figures 1B, S5G, S5H), implying that milk in knockout (KO) mice was sufficient for six pups but insufficient for litters of seven or more pups during late lactation (L15-L21). To further confirm the survival decrease after L15, we repeated the experiments with additional animals (accumulated litter number = 13 in Figure 1D in revised manuscript), and found the same pattern. We therefore speculate that pup deaths at L15 were due to “severe malnourishment”.

We agree that litter sizes should be adjusted to the same number when comparing wild-type (WT) and KO lactating animals. It is understandable that variation in litter size may not accurately reflect milk secretion ability of various mammary glands. In the previous procedure, we kept all litters intact and counted pup survival, and found decreased pup survival when using mixed litter size (initial Figure 1B). Of note, this calculation contained a large number of dams (WT mother = 49; KO mother = 47) and pups (WT pups = 342; KO pups = 316). Thus, within these numbers, litters of the same size (pup $n = 6$, $n = 7$, and $n = 8$, new Figures 1B, S5G, S5H, respectively) were re-

calculated and showed a similar pattern to that of mixed litter size. The curve with an adjusted litter size of $n = 7$ (WT mother = 18; KO mother = 17) is shown in the main figures for clarification (Figure 1B).

Thank you for highlighting the phenotype in *Btn1a1* KO mice (Ogg et al. 2004. *PNAS*). The *Tardbp* KO mice showed a similar phenomenon of rapid litter death (whole litters or a fraction of each litter). To show our data with higher resolution, we demonstrated “pup death within 2 d post-birth” classified by all dead, fraction dead, and all alive (Figure S5F). As shown, KO mice experienced much higher all dead and fraction dead rates than WT mice, suggesting lactation insufficiency in *Tardbp* KO mice.

In the second gestation, the survival rate of pups nursed by KO mice still decreased, although the rate was higher (approximately 70% in second gestation versus near 60% in first gestation). Similar to that in the first gestation, KO mice could not nurse 100% of pups during late lactation if the litter size was greater than seven (data not shown). We collected additional second gestation animals (litter size = 7), with data shown in Figure 1C.

Regarding pup accessibility to solid food, we set up a camera to monitor pup behavior from L15 to L21 continuously (data not shown). The video records indicated that pups from WT mothers occasionally ate solid food from L19, whereas pups from KO mothers could not climb up to obtain solid food until L21 if the litter size was greater than seven. This is possibly because pups from KO mothers were thinner and smaller compared to those from WT mothers.

We appreciate your careful review, which has been very helpful for improving the manuscript.

(5) Page 6, lines 13-15. It would be more reassuring if the data on apoptosis and cell division were quantified (e.g., by counting TUNEL and Ki67 positive nuclei) with sufficient replicates for statistical significance. This also applies in late lactation (Page 12, lines 18-21 and Page 13, lines 1-3, Fig. 7). In this latter case, compensatory cell division may also be occurring, which could be confirmed using Ki67 as a marker [unpublished data of this reviewer, shows both increased apoptosis and cell renewal

occurring in the *Btn1a1*^{-/-} mouse during peak lactation].

Response: Thank you for your suggestions. We have counted the TUNEL and Ki67 positive cells at P17.5 as suggested (Figures S6E, S6F). The statistical results from six mice for each genotype supported our previous conclusion that *Tardbp* KO showed indistinguishable changes in cell apoptosis and proliferation at P17.5.

During late lactation, we counted shed cells (collapse of mammary alveoli) to determine the apoptotic cells at L15 and L18 (Figure S12A). Moreover, we detected cell division using Ki67 staining during late lactation, and results indicated that cell proliferation did not change at L15 and L18, but did decrease at L21 in KO mice (Figures S12B, S12C). These data have been described in the revised manuscript (Page 14, lines 19-22).

(6) Page 6, lines 18-19, and Page 22, lines 18-22 (Fig. 2). How was milk quantitatively recovered from mice, especially at day 2 of lactation? This is technically very difficult. Was a vacuum pump used? If so, how were evaporative losses controlled? Pup weight can serve as a useful indirect way of comparing milk yields among test and control groups, so it is surprising that there is no correlation between the milk yield differences on lactation day 2 in Fig. 2 with the pup weight data in Fig. 1. Another potential problem is the amount of oxytocin used to stimulate milk let-down if the doses used were 10 international units. If so, this is at least 50-fold more than required and can precipitate exfoliation of the epithelium (especially important if these mice were used for further experiments after lactation day 2 – e.g., milking on day 10).

Response: The milk collection from lactating mice was performed according to previously published methods (Willingham et al. *J Vis Exp.* 2014. Milk collection methods for mice and Reeves' muntjac deer), with minor modification. Female lactating mice were separated from their pups at L2 or L10 for 3-5 h, anesthetized with xylazine, and injected intraperitoneally with oxytocin to induce milk let-down. After 10 min, one researcher held the anesthetized mouse while manually expressing milk, with another researcher collecting the milk using a P-200 Pipetman into a 1.5-ml tube. After transferring milk drops into the tube, the tube lid was closed immediately to minimize

liquid loss. Total time required to milk an animal was no longer than 40 min. All teats were milked in order starting with the right upper pelvic and going clockwise to the right upper axillary gland. After collection, the mice were dissected to see whether milk was completely collected, as assessed by eye and immunofluorescence staining using milk antibody. If milked incompletely, a large amount of milk could be seen within the mammary ducts, and the data on these mice were excluded from further analysis.

In our previous experiments, we applied a large amount of oxytocin (10 IU) to fully stimulate milk let-down. Although smaller dosages are well recognized for mice (0.2 IU, Ogg et al. *PNAS*. 2004), the published protocol mentioned that smaller dosages of oxytocin would not obtain enough milk at early lactation (L1) (Figure 3a in Willingham et al. *J Vis Exp*. 2014). Moreover, a large dose of oxytocin (10 IU) has been applied in previously published articles for milk quantitation (e.g., Wang et al. *Nat Med*. 2012. Cidea is an essential transcriptional coactivator regulating mammary gland secretion of milk lipids; Boxer et al. *Cell Metab*. 2006. Isoform-specific requirement for Akt1 in the developmental regulation of cellular metabolism during lactation). We therefore applied 10 IU in our experiments. Considering that a high amount of oxytocin can lead to precipitate exfoliation of the epithelium and other potential effects, we did not use these mice for further experiments. All milked mice were sacrificed to check whether milk collection was complete (if incomplete, the data on these mice were excluded from further analysis). To further omit the possible effects caused by a high amount of oxytocin, we also used a smaller dosage (0.2 IU) to re-perform this experiment. The milk volume collected from WT mice was similar to published results (Willingham et al. *J Vis Exp*. 2014). Furthermore, the new results indicated that *Tardbp* KO led to milk production deduction at L10 (Figure S7A), confirming our previous conclusions.

We agree that pup weight can serve as a useful indirect way to compare milk yields among test and control groups. In our results, pup weights between WT and KO mothers were indistinguishable on L2 (Figure 1E). One possible explanation is that, although milk production between WT and KO mothers differed during the first 2 d of lactation, the weight of pups at birth from the two genotypes did not differ at L2 (data not shown) because the weight derived from exogenous nutrition took time (longer than

2 d) to exhibit differences. As time increased, the pup weight data were correlated with the milk yield difference (Figure 1E).

We apologize for our unclear description in the initial manuscript. The related details and necessary changes have been added in the revised manuscript (Pages 25-26).

(7) Page 8 Lines 3-9 (Fig. 3). Lipid droplet size differences appear comparable (although somewhat less) with the *Btn1a1*^{-/-} mouse [Ogg, S.L. et al., *ibid*], which deserves some comment as you would expect the two mutant lines to phenocopy each other since the mRNA and protein levels of *Btn1a1* are depleted in *Tardbp*^{-/-} mice (Fig. 4B,E).

Response: Thank you for your suggestions. We have added the corresponding statement to the revised manuscript (Page 17, last paragraph).

(8) Page 9. Lines 1-3 (Fig.4). The RNAseq data are not adequately discussed. What other differences were noted related to lipid synthesis and secretion? Apparently quite a few, judging from later comments on page 15, line 15 - page 16, lines 1-3. What objective criteria were used to select *Btn1a1* and *Xdh* at the expense of other potential candidates?

Response: Thank you for the question. Among the differentially expressed genes identified from the RNA-seq data in the *Tardbp* KO epithelium (Figure 4A), there were 44 genes (expression fold change $|\log_{2}FC| > 0.7$, $P < 0.001$, and FDR < 0.01 , Table S2) involved in lipid metabolism. It has been reported that the TDP-43 protein regulates downstream genes by directly binding to the motif of UG-enriched sequences (Lukavsky et al. *Nat Struct Mol Biol.* 2013. Molecular basis of UG-rich RNA recognition by the human splicing factor TDP-43). Therefore, to identify the genes directly regulated by TDP-43, we performed statistical analysis to determine enrichment of the UG-repeated motif in the mRNA of the above 44 genes. A similar method has been applied in previous reports (e.g., Paz et al. *Nucleic Acids Res.* 2014. RBPmap: a web server for mapping binding sites of RNA-binding proteins; Piva et al. *Hum Mutat.* 2012. SpliceAid 2: A database of human splicing factors expression data

and RNA target motifs). Here, *Btn1a1* mRNA showed the most significant enrichment in the UG-repeated motif. Moreover, BTN is well-recognized to be important for milk lipid secretion, phenocopying with *Tardbp* KO mice (Figures 1 and 3). This suggests that BTN may be a downstream gene of TDP-43 for regulating milk lipid secretion.

Further qRT-PCR assay confirmed the significant decrease in *Btn1a1* expression in the *Tardbp* KO mammary gland at L1 and L10. We also found that the expression level of *Xdh*, an important interactor with BTN for milk lipid secretion (McManaman et al. *J Mammary Gland Biol Neoplasia*. 2007. Molecular determinants of milk lipid secretion), was significantly decreased at L1 and L10, whereas the expression of *Cidea* was comparable at P17.5 and L10 at both the mRNA and protein level (Figures S10A-S10C). We therefore focused on the *Btn1a1* and *Xdh* genes, which are well recognized for their mediation of milk lipid secretion.

The above details have been added to the revised manuscript (Page 9, line 14-Page 10, line 8).

(9) Page 12, lines 4-6 (Fig. 8D). Reference to this figure is out of place here and belongs in the Discussion with a broader discussion of mRNA regulation by TDP-43 in other contexts.

Response: Thank you for the suggestion. We have made the corresponding modifications (Page 16).

(10) Page 12, lines 9-11. This is one of several places where the Monks paper (see comment 2, above) should be highlighted with appropriate changes to the text.

Response: According to your suggestion, we have deleted this statement (Page 14, lines 9-10) and discussed the recommended paper in our revised manuscript (Page 18, lines 4-15).

(11) Page 13, lines 17-22, page 14, lines 1-3 (Fig. 8). It is not clear how the human subjects were followed up 7-8 weeks post-partum. Were these women formula feeding by choice or necessity? If by choice, then the data are meaningless. If by necessity, the

data may have some merit, although there are only two 100% formula feeders, which cannot approach statistical significance.

Response: We apologize for the missing information in the initial manuscript. Fresh human milk samples were collected and analyzed for mRNA expression levels. The milk samples were then divided into three groups (exclusive breastfeeding, partial breastfeeding, and formula-feeding) based on follow-up telephone interview during the 7-8-week lactation period to confirm that partial breastfeeding and formula-feeding mothers were driven by necessity.

Although the formula-feeding group appeared to show lower expression of TDP-43 relative to the other groups, there were only two cases, which was insufficient for statistical analysis. Unfortunately, we were unable to obtain more clinical cases for this group. Nevertheless, TDP-43 was significantly up-regulated in the exclusive breastfeeding group in comparison with the partial breastfeeding group, which suggests that low expression level of TDP-43 may result in lactation deficiency in human milk secretion.

Necessary statements and appropriate changes have been added to the revised paper (Pages 15, 28).

(12) Page 15, lines 9-11. The severe premature involution observed in Vorbach's Xdh+/- mouse does not agree with the more recent conditional Xdh-/- line described by Monks et al.

Response: Thank you for the suggestion. The corresponding modifications have been made (Page 18).

(13) Page 15, line 15. A more complete discussion of the functions of TDP-43 in lipid homeostasis might help put the current data in a wider context.

Response: Thank you for your constructive suggestion. A more in-depth discussion on the functions of TDP-43 in lipid homeostasis has been added to the new manuscript (Last paragraph on Page 18).

(14) Page 16, lines 13-19. Unless the data on progesterone are shown in Results, this section of the Discussion needs to be deleted. Also, progesterone levels go down at parturition, at the time that TDP-43 should be increasing, so how can it control TDP-43 during lactation?

Response: We agree with your suggestion. We have removed this part of the discussion.

(15) Fig. S4A. Lactation day 2?

Response: Yes. The above detail has been added to the Results (Page 7, line 16) and figure legends of Figure S7B.

(16) Fig. S5. Gene expression and immunofluorescence data are not proxies for the measurement of protein synthesis or secretion.

Response: Thank you for your suggestion. We have changed it to “the expression levels of several essential genes related to milk protein in MECs were unaffected by *Tardbp* loss” (Page 7, lines 17-18).

Reviewer #2 (Remarks to the Author):

In this work, Zhao et al. have investigated the role of mouse TDP-43 in facilitating milk lipid secretion by post-transcriptional regulation of two genes, *Xdh* and *Btn1a1* that are required for the secretion of lipid droplets from epithelial cells into the lumen. Following evolutionary analysis, the authors suggest that for this function TDP-43 has undergone positive selection in mammals and that knockout of the *Tardbp* gene in mice resulted in lipid droplet secretion failure associated with poor newborn survival. Finally, clinical samples from lactating women further substantial the roles of TDP-43 in milk secretion.

Overall, the results of this study could be interesting but the mRNA aspects are not fully convincing:

Response: Thank you for your positive comments and constructive suggestions.

We have made all corresponding modifications, as discussed in the following paragraphs.

1) in the evolutionary analysis, the authors identify both TARDBP and SRSF9 as undergoing positive selection. However, the possible role of SRSF9 is not further investigated, although it is an RNA binding protein like TDP-43 and could be affecting processing of the Btn1a1 and Xdh genes as well. Has this been investigated at all by the authors using a simple approach such as knocking down this factor (either alone or together with TDP-43) in HC11 cells and see whether these two genes might have a synergistic effect?. On page 4 the authors justify looking at TDP-43 because of previous work that suggest a role of this protein in breast cancer progression. However, also SRSF9 has been found to be altered in many kind of cancers and therefore in this respect there does not seem to be a big difference between the two.

Response: Thank you for your suggestions. We knocked down SRSF9 in a differentiated mammary epithelial cell line (HC11) and found that the protein levels of BTN and XOR did not change significantly. The data are shown in Figures S12D and S12E in the revised manuscript (Page 16, line 20 to Page 17, line 2).

2) In Figure 6, the authors show that TDP-43 can bind the 3'UTR of Btn1a1 and Xdh and affect the mRNA stability of the transcripts. However, the results of the experiments are not fully convincing. For example, using Flag-TDP-43 the authors show that the relative expression of Btn1a1 and Xdh can be increased following overexpression of TDP-43. However, because the authors just report a ratio, this experiment does not really show that TDP-43 can successfully recover the expression of wild-type protein levels, only that there is a rather moderate increase in expression. Has this experiment been performed?. Ideally, the authors should transfect HC11 cells with TDP-43 and show that Btn1a1 and Xdh protein expression can be fully recovered to wild-type levels. As a sidenote, using Flag is also not the ideal control for these experiments. As the authors hypothesize that TDP-43 mRNA binding is instrumental in affecting mRNA stability a much more convincing control would be a Flagged TDP-43 that carries

mutations in the RRM1 and RRM2 regions.

Response: Thank you for your suggestions. To determine whether TDP-43 could regulate *Btn1a1* and *Xdh* mRNA stability, we first detect the expression of *Btn1a1* and *Xdh* at both the mRNA and protein level from WT and KO mice *in vivo*, and found that *Tardbp* KO significantly decreased the expression of *Btn1a1* and *Xdh* at both the RNA and protein level (Figures 4B-4E). We next employed shRNA-targeted *Tardbp* in differentiated HC11 cells *in vitro* and found a marked reduction in *Btn1a1* and *Xdh* expression upon TDP-43 knockdown (Figures 4F). To confirm whether TDP-43 could positively regulate *Btn1a1* and *Xdh* protein expression, we also redesigned a gain-of-function assay. Western blot assay showed a marked increase in BTN and XOR protein levels upon TDP-43 overexpression compared with the control group (Figure S11A). From both loss-of-function and gain-of-function assays, we found that TDP-43 positively regulated BTN and XOR expression.

In addition, we redesigned the experiment to detect *Btn1a1* and *Xdh* mRNA stability after overexpression of TDP-43. *Btn1a1*, *Xdh*, and *Gapdh* mRNA levels were normalized to 18S rRNA between the control and TDP-43 overexpression group at time 0 (before actinomycin D addition) or 2, 4, 6, and 8 h after actinomycin D addition. The half-lives ($t_{1/2}$) were calculated as the time of each mRNA to decrease to 50% of its initial abundance. Results showed no difference in *Gapdh* mRNA stability between the two groups, whereas both *Btn1a1* and *Xdh* mRNA became more stable upon overexpression of TDP-43 in comparison with the control (Figure 6C), which is consistent with our previous view.

To confirm that TDP-43 mRNA binding was instrumental in affecting mRNA stability, we also detected *Btn1a1* and *Xdh* mRNA stability after overexpressing the TDP-43 deletion mutation (C-term) without RNA binding domains (RRM1 and RRM2). As expected, overexpressed TDP-43 full length (Flag-FL), but not TDP-43 C-terminal domain, significantly suppressed the decay of *Btn1a1* and *Xdh* mRNA in the differentiated HC11 cell line (Figure 6C).

These results have been added to the revised manuscript (Pages 12-13).

3) In the 293T cells transfected with the GFP constructs carrying the 3' UTR of the *Btn1a1* mRNAs, Figure 6, a critical control that is lacking and is represented by a control where the UG-rich regions in this 3'UTR have been deleted. One would then expect that the regulatory property of TDP-43 would be lost. Secondly, why was this reporter experiment also not repeated for the 3'UTR of *Xdh*?

Response: Thank you. According to your suggestions, we redesigned and performed the GFP reporter assays, which supported our previous conclusions.

To demonstrate that TDP-43 regulated *Btn1a1* mRNA stability through the 3' UTR, we inserted the 3' UTRs of *Btn1a1* or *Xdh* following the green fluorescent protein (GFP) gene into a mammalian expressive vector, representing GFP-*Btn1a1*-UTR and GFP-*Xdh*-UTR, respectively (Figure 6D, upper), and then transfected the cells to determine GFP change after knockdown of TDP-43. As expected, we found that knockdown of TDP-43 led to substantially lower GFP expression in the GFP-*Btn1a1*-UTR and GFP-*Xdh*-UTR groups compared with that in the scramble control (Figure 6D).

We next generated deletion mutations of the *Btn1a1* and *Xdh* 3' UTRs without TDP-43-binding sites (GFP-*Btn1a1*-mutUTR and GFP-*Xdh*-mutUTR, respectively). RIP-qRT-PCR assay was employed to confirm that deletion mutations of UG-enriched sequences severely abated the interaction between GFP mRNA and TDP-43 protein (Figure S11B). We then detected the GFP expression of mutant vectors upon sh-TDP-43 knockdown and found that mutation of the TDP-43 binding site completely abolished the negative regulation of GFP expression upon TDP-43 knockdown (Figure 6D). To measure whether TDP-43 could regulate the mRNA stability of the above GFP reporters, we also detected GFP mRNA stability of each group upon actinomycin D treatment after TDP-43 knockdown. Results showed that GFP mRNA stability of GFP-*Btn1a1*-UTR and GFP-*Xdh*-UTR, but not GFP-*Btn1a1*-mutUTR or GFP-*Xdh*-mutUTR, significantly decreased upon sh-TDP-43 treatment compared with the control group (Figure S11C).

These results have been added to the revised manuscript (Pages 13-14).

4) Finally, are the UG-repeated sequences in the human *Btn1a1* and *Xdh* 3'UTRs

conserved with respect to the mouse genes?. If not, then the connection between the mouse data and the milk yield in post-partum women shown in Figure 7 may not be very compelling. Even if they are conserved, the authors should show that the human genes are also positively regulated by human TDP-43 in the same way as the mouse genes

Response: The regulation of *Btn1a1* and *Xdh* by TDP-43 is conserved between mice and humans: (1) the TDP-43 protein sequence is highly conserved between humans and mice, and both of their TDP-43 proteins bind to the UG-enriched motif (e.g., Kuo et al. *Nucleic Acids Res.* 2014. The crystal structure of TDP-43 RRM1-DNA complex reveals the specific recognition for UG- and TG-rich nucleic acids; Wang et al. *Genomics.* 2004. Structural diversity and functional implications of the eukaryotic TDP gene family; Lukavsky et al. 2013. *Nat Struct Mol Biol.* Molecular basis of UG-rich RNA recognition by the human splicing factor TDP-43); and, (2) UG-enriched motifs exist in both human *BTN1A1* and *XDH* mRNA sequences. To determine whether TDP-43 could regulate BTN and XOR expression in humans, we introduced human *BTN1A1* and *XDH* 3' UTRs into GFP reporters to generate GFP-h*BTN1A1*-UTR and GFP-h*XDH*-UTR, respectively. Immunoblot analysis showed that knockdown of TDP-43 reduced GFP expression of both GFP-h*BTN1A1*-UTR and GFP-h*XDH*-UTR in comparison to the sh-Control group (Figure S11D).

These results have been added to the revised manuscript (Page 14). We thank you for your constructive suggestions, which have helped improve the quality of this paper.

Reviewer #3 (Remarks to the Author):

The authors have identified the RNA-binding protein TDP-43 as a major regulator of milk production in mice and provide evidence that *Btn1a1* and *Xdh* 1 mRNAs are effectors of this phenotype. The overall findings of the report are novel and interesting. While the molecular players were previously known to participate in mRNA regulation (TDP-43), lipid homeostasis (*BTN1A1*), the implication of *XDH1* in milk production is new. The work generally supports the model proposed by the authors, but several

molecular specifics need to be strengthened and completed.

Response: Thank you for your positive comments. We agree with your suggestions, which have greatly helped improve the quality of our manuscript.

1. In Figures 5B, 5C, and 5H, the customary presentation of RIP data is as 'relative enrichment' of a given mRNA. The levels of that mRNA in IgG IP are set as '1' and the relative levels of the same mRNA in TDP-43 IP are shown as 'fold' difference relative to '1'.

Response: Thank you for your suggestion. We have revised it accordingly (Figures 5B, 5C, and 5H).

2. Throughout the figures the authors should use the same nomenclature they use in the text: lowercase italics *Btn1a1* and *Xdh1* when referring to mouse gene, mRNA and fragments of mRNA. This nomenclature is not used correctly in the figures, although it is overall used correctly in the text.

Response: Thank you for your suggestion. We have checked and made the necessary modifications to our previous draft.

3. In Figure 6, the representation of stability data also needs revision. These data should include decay rates for *Gapdh*, *Btn1a1* and *Xdh1* mRNAs with the following modifications:

- Control groups (before silencing) should be plotted for each mRNA in order to know their intrinsic stability before silencing TDP-43.

- Silencing groups should be plotted for each mRNA in order to assess their stability after silencing TDP-43. Typically, for a given mRNA the control siRNA group and the specific siRNA group are plotted in the same graph.

- Plotting should be done on a semilog scale (log Y axis, linear X axis).

- Half-lives should be determined for each mRNA as they cross the 50% line (t1/2)

Response: According to your constructive suggestions, we have clarified these issues in the revised manuscript (Figures 6A-6C and S11C).

4. In Figure 6, the authors must complete the analysis of stability determinants in *Btn1a1* and *Xdh1* 3'UTRs. They need to prepare and test reporter constructs in which the full-length 3'UTR for each mRNA is assayed as well as reporters in which the major RNA segments in the 3'UTRs found to interact with TDP-43 (as found in Figure 5) are mutated. They then should assay if wild-type and mutant 3'UTRs display changes in (1) TDP-43 binding and (2) the stability of each reporter mRNA.

Response: Thank you for your suggestions. We have redesigned and performed the GFP reporter assays accordingly, which has made our previous conclusions more robust and convincing.

To further demonstrate that TDP-43 regulated *Btn1a1* mRNA stability through the 3' UTR, we inserted the 3' UTRs of *Btn1a1* and *Xdh* following the green fluorescent protein (GFP) gene into a mammalian expressive vector, representing GFP-*Btn1a1*-UTR and GFP-*Xdh*-UTR, respectively (Figure 6D, upper), and then transfected the cells to determine GFP change after knockdown of TDP-43. RIP-qRT-PCR assays were then conducted, which showed strong interaction between GFP-*Btn1a1*-UTR/GFP-*Xdh*-UTR and the TDP-43 protein (Figure S11B). To confirm that UG-enriched motifs mediated the interaction between GFP-*Btn1a1*-UTR/GFP-*Xdh*-UTR and the TDP-43 protein, we next generated deletion mutations of the *Btn1a1* and *Xdh* 3' UTRs without TDP-43-binding sites (GFP-*Btn1a1*-mutUTR and GFP-*Xdh*-mutUTR, respectively). As expected, deletion mutations of UG-enriched sequences largely abated the interaction between the *Btn1a1/Xdh* 3' UTRs and TDP-43 (Figure S11B). To confirm the regulation of *Btn1a1* and *Xdh* mRNA stability by TDP-43, we detected the GFP mRNA stability upon actinomycin D treatment after TDP-43 knockdown. Results showed that GFP mRNA stability of GFP-*Btn1a1*-UTR and GFP-*Xdh*-UTR, but not GFP-*Btn1a1*-mutUTR or GFP-*Xdh*-mutUTR, significantly decreased upon sh-TDP-43 treatment compared with the control group (Figure S11C). Moreover, to measure whether TDP-43 could regulate GFP levels of the above reporters, we employed Western blot analysis to measure GFP expression of each group, and found that knockdown of TDP-43 led to substantially lower GFP expression in the GFP-*Btn1a1*-

UTR and GFP-*Xdh*-UTR groups compared with that in the scramble control (Figure 6D), whereas GFP expression remained unchanged upon sh-TDP-43 knockdown in the GFP-Control-UTR group. Furthermore, mutation of the TDP-43 binding site completely abolished the negative regulation of GFP expression upon TDP-43 knockdown (Figure 6D).

These results have been added to the revised manuscript (Pages 13-14).

5. In Figure 7, can the milk production phenotype rescued in the *Tardbp*^{-/-} mice if *Btn1a1* and *Xdh 1* mRNAs are re-expressed? For example, the authors could deliver via lentivirus *Btn1a1* and *Xdh 1* mRNAs that lack the 3'UTR and hence are intrinsically stable. This experiment would provide critical support to the authors' model.

Response: Thank you for your suggestions. In our manuscript, knockout of TDP-43 in the mouse mammary gland affected mRNA stability of *Xdh* and *Btn1a1* and led to the down-regulation of milk production. Ideally, if XOR and BTN proteins were stably re-expressed, the milk production phenotype should be rescued in *Tardbp*^{-/-} mice.

We generated lentiviruses of *Btn1a1* and *Xdh* mRNAs, which lacked 3' UTRs, and then injected them into WT and TDP-43 KO mice, respectively. However, we did not observe significant differences in milk production between WT and KO mice. This was likely due to the following possible reasons: 1) the inflammatory response caused by lentivirus delivery led to breast mastitis (Wellenberg et al. *Vet Microbiol.* 2002. Viral infections and bovine mastitis: a review), which severely impacted milk production. In fact, the mammary gland in our experiments was found to be filled with milk and other material, including bacteria, which supports the previous statement; and 2) lentivirus infection efficiency of lactation mammary cell *in vivo* was not enough to demonstrate differences in milk production. Although we observed positive signals (GFP) of *Xdh* and *Btn1a1* after injection, the infection efficiency was relatively low. High-titer lentiviruses are reportedly required for infecting primary mammary epithelial cells *in vitro* (Welm et al., *Cell Stem Cell*, 2008. Lentiviral transduction of mammary stem cells for analysis of gene function during development and cancer), and a high concentration of milk proteins, such as lactadherin and peptides, within the mammary ducts is known

to adsorb and interfere with lentivirus infection (Arnold et al. *Antiviral Res.* 2002. Antiadenovirus activity of milk proteins: lactoferrin prevents viral infection). The proportion of cells transduced by adenoviruses is also reported to be low (Intraductal injection into the mouse mammary gland, p. 259-270. In M. M. Ip and B. B. Asch (ed.), *Methods in mammary gland biology and breast cancer research*). Thus, delivery of exogenous viruses may be unsuitable for phenotype rescue of milk production. To the best of our knowledge, there are no reports on lentivirus delivery for studies on milk production in mouse mammary glands *in vivo*.

Nevertheless, we totally agree with your suggestions that functional phenotype rescue would critically support our working model. Therefore, we made some attempts on HC11 cells to further validate our conclusions. The HC11 cells are a mammary epithelial cell line derived from mid-pregnant Balb/c mice (Ball et al. *EMBO J.* 1988. Prolactin regulation of β -casein gene expression and of a cytosolic 120-kd protein in a cloned mouse mammary epithelial cell line). The differentiated HC11 cells upon lactogenic hormone mix (dexamethasone, insulin, and prolactin) form blister-like structures, called domes, which are believed to result from fluid secretion by mammary epithelial cells. We first measured dome formation upon sh-TDP-43 knockdown in response to treatment with lactogenic hormones and found that knockdown of TDP-43 in HC11 cells impaired dome formation (Figure S10D). When co-expressed with *Btn1a1* and *Xdh1* coding sequences in TDP-43-knockdown cells, we found that the decrease in dome number by sh-TDP-43 knockdown was partially rescued (Figures S10E and S10F).

These results and methods have been added to the revised manuscript (Page 10).

6. Although the text reads well, some attention to grammar, syntax, and word choice is needed. One example is the authors' use of the word 'imply', when they mean instead 'suggest' or 'indicate'.

Response: Thank you for your suggestions. The revised manuscript has been edited and proofread by a native English speaker.

Reviewer #4 (Remarks to the Author):

The authors analyzed the role of RNA-binding protein (RBP) TDP-43 in milk lipid secretion in mammals. Most of the manuscript concerns functional and molecular analyses of this RBP and its encoding gene, in particular using a KO mouse model. These analyses are far from my expertise and will be carefully addressed by other reviewers. I concentrate my analysis of the manuscript on its (very) short first part (pp 3-4), which deals with evolutionary analysis of RBPs in mammals. This analysis led the authors to conclude that TDP-43 underwent positive selection in mammals and this conclusion is presented as a main reason to further study the role of this particular RBP in milk lipid production.

There are two problems with this section of the manuscript.

First is about the way this analysis is presented. I don't think that the authors can claim, based on their very limited analysis, that « we performed the evolutionary analysis to study the molecular mechanism of milk secretion » (page 3 lines 6-7). Not sure neither that they can claim this analysis allowed to identify « the essential post-transcriptional regulator in mammals » (page 3 line 18). I think that the authors should explain in a more convincing way why this analysis is useful and why genes undergoing positive selection could be good candidate genes involved in lactation in mammals.

Response: Thank you for your suggestions, which have been very helpful for improving our article. We have systematically and thoroughly revised our manuscript, including the section related to positive selection analyses on RBPs.

Lactation is considered one of the most characteristic features in mammalian species. Milk can nourish neonates and help establish immunological and endocrine competence in offspring, showing a survival advantage to eggs of ancestral species (Capuco and Akers. *J Biol.* 2009. The origin and evolution of lactation). Thus, the features of mammalian lactation have been accrued gradually through radiations of synapsid ancestors by natural selection (Oftedal OT. *J Mammary Gland Biol Neoplasia*, 2002). The mammary gland has been regarded as an organ under “screening force” by natural selection in both milk secretory processes (Darwin C 1872. *On the origin of*

species by means of natural selection, or the preservation of favored races in the struggle of life. John Murray, London, UK) and secretory structures/milk components (e.g., Oftedal OT. The mammary gland and its origin during synapsid evolution. *J Mammary Gland Biol Neoplasia*, 2002; Lefevre et al. *Annu Rev Genomics Hum Genet*, 2010. Evolution of lactation: ancient origin and extreme adaptations of the lactation system).

Hundreds of genes involved in lactation evolution have been identified (e.g., Lemay et al. The bovine lactation genome: insights into the evolution of mammalian milk. *Genome Biology*, 2009; Oftedal OT. The evolution of milk secretion and its ancient origins. *Animal*, 2012); however, lactation-involved genes under natural selection remain little known. Positive selection, compared to purifying selection, is an important force of natural selection (Yang et al. Statistical methods for detecting molecular adaptation. *Trends Ecology and Evolution*, 2000), which indicates that those genes responsible for lactation regulation have been positively selected in mammals during evolution. Lactation-related genes can be identified by calculating positive selection signals in the ancestral branches of mammals, as positive selection genes in these ancestral branches are more likely associated with characteristics of mammals versus other animals (fish, birds, reptiles). Thus, we performed a phylogenomic analysis of positive selection in 15 vertebrate genomes to screen candidates for lactation.

For the “positive selection” approach, previously published research has demonstrated its usefulness in screening candidate genes. For example, a previous genomic scan of positively selected genes among one-to-one orthologous genes in the tree shrew (*Tupaia belangeri chinensis*) genome and five other phylogenetically closely related mammals identified several genes under positive selection, including *trpv1*, which may be related to spiciness tolerance in tree shrews. Further functional analyses confirmed that transient receptor potential vanilloid type-1 (TRPV1) ion channel (tsV1) lowered sensitivity to capsaicinoids (Han et al. *PLoS Biol*, 2018. Molecular mechanism of the tree shrew's insensitivity to spiciness).

Moreover, we performed whole-genome scans for positive selection genes, which showed that *XDH*, a well-known milk secretion gene, was top-ranked in the list of

positive selection genes (the second most robust among 7431 one-to-one orthologous genes, data not shown). This suggests the feasibility of using positive selection to screen candidate genes involved in lactation. In the current study, our subsequent functional experiment in *Tardbp* KO mice demonstrated that TDP-43 is essential for milk lipid secretion and pup growth during lactation, further confirming our conclusions.

Several reports have suggested that regulation of RNA stability or regulation at the post-transcriptional level may be the key to lactation activation (Lemay et al. *BMC Syst Biol.* 2007. Gene regulatory networks in lactation: identification of global principles using bioinformatics; Vander et al. *Reprod Fertil Dev.* 2015. Gene expression in the mammary gland of the tammar wallaby during the lactation cycle reveals conserved mechanisms regulating mammalian lactation). However, to the best of our knowledge, little information is currently available on the functional roles of regulators on lactation at the post-transcriptional level. As RNA-binding proteins (RBPs) mediate key steps in post-transcriptional regulation of gene expression (Gerstberger et al. *Nat Rev Genet.* 2014. A census of human RNA-binding proteins; Harvey et al. *Biochem Soc Trans.* 2017 Post-transcriptional control of gene expression following stress: the role of RNA-binding proteins), we focused on screening candidate genes within RBPs for the regulation of lactation.

We apologize for our unclear description in the initial manuscript. The relative details and necessary changes have been added to the revised manuscript (Page 3, line 17 to Page 4, line 16; Page 16, line 16 to Page 17, line 2; Pages 20-21).

Second problem is that from what is shown in the manuscript and supplementary data, it is not very clear what has been done to select the gene families for which dN/dS has been analyzed. It should be explained what is the « stringent filtering » (page 3 line 20) that has been used. It is not clearly explained what is reported in the supplementary file in columns « Model A alternative » and « Model A null » and the calculated dN/dS values are not shown anywhere. At least for the two gene families on which emphasis is put (TARDBP and SRSF9), multiple alignments should be provided as supplementary figures.

Response: Thank you. We have added detailed description of the stringent filtering and models in the Materials and Methods section (Page 21), as well as the dN/dS values in the main text (Page 4, lines 13-14, Table 1) and multiple alignments of TDP-43 and SRSF9 in the Supplementary Information (Figures S2, S3).

Reviewers' comments:

Reviewer #1 (Remarks to the Author):

This manuscript is much improved from the original submission. A few minor suggestions and corrections:

p. 5, line 18. "Gestation" refers to development in utero, so you can't have a pup survival rate until after birth. Suggest ...pup survival rate improved in the KO group during the second lactation...

p. 6, line 20. This statement is ambiguous. Suggest Induced failure of both pup viability and growth...

p. 7, lines 11-13. Figures 2A and S7A. Emphasize why two concentrations of oxytocin were used, i.e., the higher dose to remove all milk at an early stage and the lower dose at mid-lactation. The fact that Willingham et al. (2014) and Wang et al. (2012) used 10 IU does not supplant the more conventional use of much lower doses for removing milk during an established lactation.

Figures 6D and S11D. In the interests of clarity, "sh-TDP-43-1 and sh-TDP-43-2" should be defined in the legend and cited in the text.

p. 14, line 18. Why is this result remarkable when it is the natural consequence of reduced milk availability?

p. 17, line 1. Figures S12D and S12E should be first cited and discussed in the Results section.

p. 18, line 3. ...maintain a certain number of pups (litter size) during late lactation...

p. 26, lines 7- 8. According to the author's response to this reviewers concerns, mice that retained milk within the mammary gland (determined by surgical dissection) were not included in the data. However, this reassurance is missing from the text.

Reviewer #2 (Remarks to the Author):

Although the authors have clarified a few issues and performed the requested experiments, some of the results are still not very convincing. In particular, in Figure 6 the authors have made the requested GFP-UTR constructs to check whether TDP-43 binding to hBTN1A1 and hXDH 3'UTRs could mediate stability. First, the results of this assay have not been subject to any apparent quantification. Second, it is not clear why sh-TDP-43-2 was used in the assay considering that its apparent ability to downregulate TDP-43 expression is much lower than the other shRNA used: sh-TDP-43-2. Most importantly, with the exception of a few lanes the levels of GFP expression do not seem to be prominently affected by TDP-43 knockout. As a result, results of these experiments remain quite unconvincing.

Reviewer #3 (Remarks to the Author):

I appreciate the authors' careful and thorough responses to my concerns. I have no further requests.

Reviewer #4 (Remarks to the Author):

none

Reviewer #1 (Remarks to the Author):

This manuscript is much improved from the original submission. A few minor suggestions and corrections:

p. 5, line 18. “Gestation” refers to development in utero, so you can’t have a pup survival rate until after birth. Suggestpup survival rate improved in the KO group during the second lactation...

Response: Thank you for your careful reading and correction. We have changed the “gestation” to “lactation” accordingly in the new text (p. 5, line 22).

p. 6, line 20. This statement is ambiguous. Suggest Induced failure of both pup viability and growth...

Response: Thank you for your suggestion. We have changed the sentence accordingly in the new text (p. 7, lines 1-2).

p. 7, lines 11-13. Figures 2A and S7A. Emphasize why two concentrations of oxytocin were used, i.e., the higher dose to remove all milk at an early stage and the lower dose at mid-lactation. The fact that Willingham et al. (2014) and Wang et al. (2012) used 10 IU does not supplant the more conventional use of much lower doses for removing milk during an established lactation.

Response: Thank you for your suggestion. The description has been added to emphasize the rationale of using two dosage of oxytocin in the new text (p. 7, lines 17-18).

Figures 6D and S11D. In the interests of clarity, “sh-TDP-43-1 and sh-TDP-43-2” should be defined in the legend and cited in the text.

Response: Thank you for your suggestion. The description has been added accordingly in the new text and figure legends (p. 13, line 17; p. 14, line 14; p. 38, lines 11-12 in Article file, and p. 5, lines 16-17 in Supplementary Information file).

p. 14, line 18. Why is this result remarkable when it is the natural consequence of reduced milk availability?

Response: Thank you for your careful reading and correction. We have deleted the “remarkable” accordingly in the new text (p. 15, line 6).

p. 17, line 1. Figures S12D and S12E should be first cited and discussed in the Results section.

Response: Thank you for your suggestion. We have cited the SRSF9 results at the beginning of functional assays (p. 5, lines 1-3).

p. 18, line 3. ...maintain a certain number of pups (litter size) during late lactation...

Response: Thank you for your careful reading and correction. We have changed the sentence accordingly in the new text (p. 18, lines 16-17).

p 26, lines 7- 8. According to the author’s response to this reviewers concerns, mice that retained milk within the mammary gland (determined by surgical dissection) were not included in the data. However, this reassurance is missing from the text.

Response: Thank you for your suggestion. The description has been added to accordingly in the new text (p. 27, lines 3-4).

Reviewer #2 (Remarks to the Author):

Although the authors have clarified a few issues and performed the requested experiments, some of the results are still not very convincing. In particular, in Figure 6 the authors have made the requested GFP-UTR constructs to check whether TDP-43 binding to hBTN1A1 and hXDH 3'UTRs could mediate stability. First, the results of this assay have not been subject to any apparent quantification. Second, it is not clear why sh-TDP-43-2 was used in the assay considering that its apparent ability to downregulate TDP-43 expression is much lower than the other shRNA used: sh-TDP-

43-2. Most importantly, with the exception of a few lanes the levels of GFP expression do not seem to be prominently affected by TDP-43 knockout. As a result, results of these experiments remain quite unconvincing.

Response: Thank you for your suggestions to improve the manuscript. For the experiments using GFP-UTR constructs to check whether TDP-43 stabilize h*BTN1A1* and h*XDH* 3'UTRs (Supplementary Figure S11D for Figure 6), we agree that one of shRNAs (sh-TDP-43-2) is not a perfect one for downregulating TDP-43 levels. We therefore re-designed the third shRNA (sh-TDP-43-3, sequence shown in Table S3) to obtain the better knockdown efficiency, and the experiments were re-performed using all three shRNAs (Figure S11D). The new results keep the same pattern with that before. The new shRNA (sh-TDP-43-3) together with sh-TDP-43-1 could downregulate TDP-43 with better knockdown efficiency, and the corresponding GFP levels also showed clear reduction. The quantification were performed from three repeats of Western blotting, which showed that knockdown of TDP-43 could largely downregulate GFP protein expression of h*BTN1A1* and h*XDH* 3'UTRs reporters (Figure S11E).

REVIEWERS' COMMENTS:

Reviewer #2 (Remarks to the Author):

Authors have added additional experiments that have made their manuscript more convincing

REVIEWERS' COMMENTS:

Reviewer #2 (Remarks to the Author):

Authors have added additional experiments that have made their manuscript more convincing

Response: We thank this reviewer for the positive comment.